# Context-invariant beliefs are supported by dynamic reconfiguration of single unit functional connectivity in prefrontal cortex of male macaques

Jean-Paul Noel [1,2,4] ✉, Edoardo Balzani [1,3,4], Cristina Savin [1,5] & Dora E. Angelaki[1,5]

Natural behaviors occur in closed action-perception loops and are supported by dynamic and flexible beliefs abstracted away from our immediate sensory milieu. How this real-world flexibility is instantiated in neural circuits remains unknown. Here, we have male macaques navigate in a virtual environment by primarily leveraging sensory (optic flow) signals, or by more heavily relying on acquired internal models. We record single-unit spiking activity simultaneously from the dorsomedial superior temporal area (MSTd), parietal area 7a, and the dorso-lateral prefrontal cortex (dlPFC). Results show that while animals were able to maintain adaptive task-relevant beliefs regardless of sensory context, the fine-grain statistical dependencies between neurons, particularly in 7a and dlPFC, dynamically remapped with the changing computational demands. In dlPFC, but not 7a, destroying these statistical dependencies abolished the area's ability for cross-context decoding. Lastly, correlational analyses suggested that the more unit-to-unit couplings remapped in dlPFC, and the less they did so in MSTd, the less were population codes and behavior impacted by the loss of sensory evidence. We conclude that dynamic functional connectivity between neurons in prefrontal cortex maintain a stable population code and context-invariant beliefs during naturalistic behavior.

Adaptive behaviors require flexible computations abstracting away task features and goals from our immediate sensory context. This ability allows identical sensory input to yield different motor outputs under distinct contexts (i.e., context-dependence). Or conversely, for a single task to be accomplished under vastly different sensory conditions (i.e., context-invariance). We may, for instance, navigate to our colleague's office in broad daylight by leveraging visual observations (e.g., optic flow), or toward our fridge in complete darkness by relying on acquired internal models – e.g., a cognitive map[1] of our home, the size of our stride. The

mechanisms subserving these flexible computations are not fully understood.

Two broad frameworks for context-dependent computation have been proposed. First, brain-wide neuroimaging studies in humans[2–4] have argued for "flexible hubs". This work has highlighted the malleable nature of functional connectivity patterns, and the fact that these patterns change with task demands. This framework has primarily emphasized the role of the fronto-parietal network, with functional connectivity within this network being (i) the most flexible, and (ii) sufficient to identify which task is being undertaken[5]. The second

[1]Center for Neural Science, New York University, New York City, NY, USA. [2]Present address: Department of Neuroscience, University of Minnesota, Minneapolis, MN, USA. [3]Present address: Flatiron Institute, Simons Foundation, New York, NY, USA. [4]These authors contributed equally: Jean-Paul Noel, Edoardo Balzani. [5]These authors jointly supervised this work: Cristina Savin, Dora E. Angelaki. ✉e-mail: Jpn5@nyu.edu

approach has argued for a dynamical systems understanding. Namely, neurophysiological recordings in macaques[6–9] coupled with population-level analyses have suggested that task context may either mold neural dynamics themselves[6], or the initial condition of neural trajectories within a fixed dynamical system[7]. Of course, these frameworks are likely not entirely independent, with theory[10–12] and nascent empirical observations[13–15] suggesting that the statistical dependencies between neural responses (i.e., functional connectivity) facilitate context-invariant decoding, as well as restrict spiking activity to a low-dimensional[16] and differentiable[17] space.

The study of neural co-fluctuations and their relation to population dynamics and flexible behavior has, however, focused primarily on the representation of simple and static stimuli[18], and on stereotyped behaviors[13–15,18,19]. Moreover, it has mostly examined a single neural node at a time[18–20], and commonly relied on measurement techniques (e.g., fMRI, calcium imaging) that allow for sampling large neural populations at the expense of temporal resolution[13–15,21]. Further, the relation between single-cell functional connectivity and population dynamics supporting flexible behavior has received more theoretical than empirical evidence (see refs. [6,7,9,22–24] for work employing artificial RNNs as "proof-of-principle" but less biological evidence). This is likely due to the difficulty in statistically quantifying unit-to-unit couplings in physiological data. Most vexingly, we do not understand how inter-neuronal correlations shape the dynamics supporting naturalistic behaviors; those unfolding over a protracted time period, within closed action-perception loops, and reliant not only on static percepts, but on dynamic beliefs regarding latent variables.

To close these gaps in knowledge, here we record single cells in sensory, parietal, and frontal cortices while leveraging two key innovations. First, we developed a closed-loop task wherein macaques navigate in virtual reality and stop at the location of briefly visible targets[25] – much akin to "catching fireflies." Successful completion of this task requires animals to dynamically update their beliefs over a latent variable: their evolving distance to the (invisible) target. Importantly, this naturalistic task allows for rapid generalization[26], and thus while animals were trained to path integrate by primarily accumulating optic flow signals ("high-density condition", Fig. 1A), they may also be tasked with navigating in a visually impoverished environment ("low-density condition", a 50-fold decrease in optic flow density, see refs. [27,28]). Normatively, in this latter context, animals had to substantially rely on an acquired internal model mapping joystick position to velocity in virtual space. Thus, animals performed a single task under two distinct computational contexts defined by the relative weighting between sensory information and an acquired internal model (see refs. [27,28] for evidence in untrained human observers that a reduction in density in optic flow results in stronger reliance on a well-established slow speed prior).

The second innovation is technical. In natural behaviors no two trials are alike, and thus we cannot compute traditional noise correlations[29] by averaging and extracting residuals. To address this challenge, we developed a Poisson Generalized Additive Model (P-GAM)[30] to account for stimuli-driven responses (i.e., tuning functions, or "signal-correlations") and estimate time-dependent and signal-independent coupling filters between neurons (i.e., the likelihood of a neuron firing given the firing of other neurons). Critically, our estimates include confidence intervals for all parameters and thus allow for individually determining their significance in explaining data (see refs. [31,32] for a similar non-Bayesian approach without the ability for inferential statistics). The overall (i.e., integrated over time) strength in coupling between two units is akin to a traditional noise correlation metric, in that signal-independent co-fluctuations are indexed. Establishing time-resolved coupling filters allows us to additionally examine the finer grain dynamics in statistical co-fluctuations between neurons. This technique recapitulates known properties of noise correlations,

such as its dependency on the distance between neurons[33] and tuning similarity[20,34] (Fig. S1, also see ref. [30] and Fig. S2 for further model validation, and REF[35] for a detailed description of spiking responses and spike-LFP phase coupling during optic-flow guided navigation, i.e., the "high-density" condition).

Our results demonstrate that a decrease in the relative weighting of sensory evidence vis-à-vis an acquired internal model during navigation is accompanied by a strengthening of relatively fixed coupling filters in sensory cortex (i.e., a gain modulation) and a dynamic reconfiguration of the finer-grain, time-resolved coupling in parietal and frontal cortices (i.e., remapping). In contrast, population dynamics were context-invariant in parietal and prefrontal cortices, but not in sensory cortex. Critically, the more the fine-grain statistical co-fluctuations reconfigured across contexts in prefrontal cortex, but not in parietal cortex, the more stable were its task-relevant population dynamics and the less was the animals' performance impacted by the loss of sensory evidence. Highlighting the role of noise correlations in engendering a stable task-relevant population code in prefrontal cortex, destroying the statistical dependencies between neurons in this area, but not in parietal or sensory cortices, eliminated this population's ability for cross-context decoding of the key latent variable, distance-to-target.

## Results

### A single closed-loop behavior under different computational demands

Prior work has studied the single neuron correlates of flexible behavior by having macaques alternate between two different tasks (e.g., color or motion discrimination[6,36]). Here, instead, we employ a single task – navigate to target – that may be accomplished under different computational constraints. We have three male macaques (Q, S, and M) use a joystick (linear and angular velocity control) to path integrate in virtual reality and stop at the location of a briefly presented (300 ms) target (Fig. 1A). Across trials, targets were randomly distributed within a radial (1–4 m) and angular (-45° to 45°) range (uniform distributions, independently sampled). The virtual environment was solely composed of ground-plane elements, which flickered (250 ms lifetime) and thus provided an optic flow signal. On a trial-by-trial basis we manipulated the density of these ground-plane elements, which were either in a "high-density" (5.0 elements/m²) or "low-density" (0.1 elements/m²) configuration (50-fold change).

Task performance was first quantified by expressing the monkeys' trajectory endpoints and target locations in polar coordinates, with an eccentricity from straight-ahead (θ) and a radial distance (r, Fig. 1A, rightmost). Figure 1B shows radial (top; r vs. r~ gain = 0.90; $R^2$ = 0.55) and angular (bottom; θ vs. θ~ gain = 0.95; $R^2$ = 0.78) responses as a function of target location for an example session (linear regression r~ ~ gain x r + intercept, see Methods and REF[25–28,35,37]). Most importantly, across all animals (n = 3) and sessions (n = 82), macaques were accurate in navigating to targets, both during high- (gains, mean ± S.E.M., radial = 0.85 ± 0.04; angular = 0.82 ± 0.14) and low-density conditions (radial = 0.79 ± 0.04; angular = 0.77 ± 0.12; contrast of high vs. low density separately for each animal and displacement dimension, all p > 0.13, all Cohen's d < 0.19, Fig. 1C, see "Behavioral Analyses" in "Methods" section). Thus, macaques were able to adaptively estimate their evolving distance to target regardless of the density of optic flow elements and the required computation.

While the location of mean endpoints did not (statistically) change across densities (Fig. 1B, C), the variance of these endpoints did (distance to target S.E.M. computed on each session, high density, 0.19 ± 0.04; low density, 0.25 ± 0.04; t(81) = 1.06, p = 0.044, Cohen's d = 0.17, Fig. 1D). This resulted in a mean decrease of 4.41% of rewarded trials in the low-density condition relative to the high-density condition (fraction trials rewarded, high-density, 0.60 ± 0.01; low-density, 0.55 ± 0.01). More importantly, under a Bayesian framework, a change

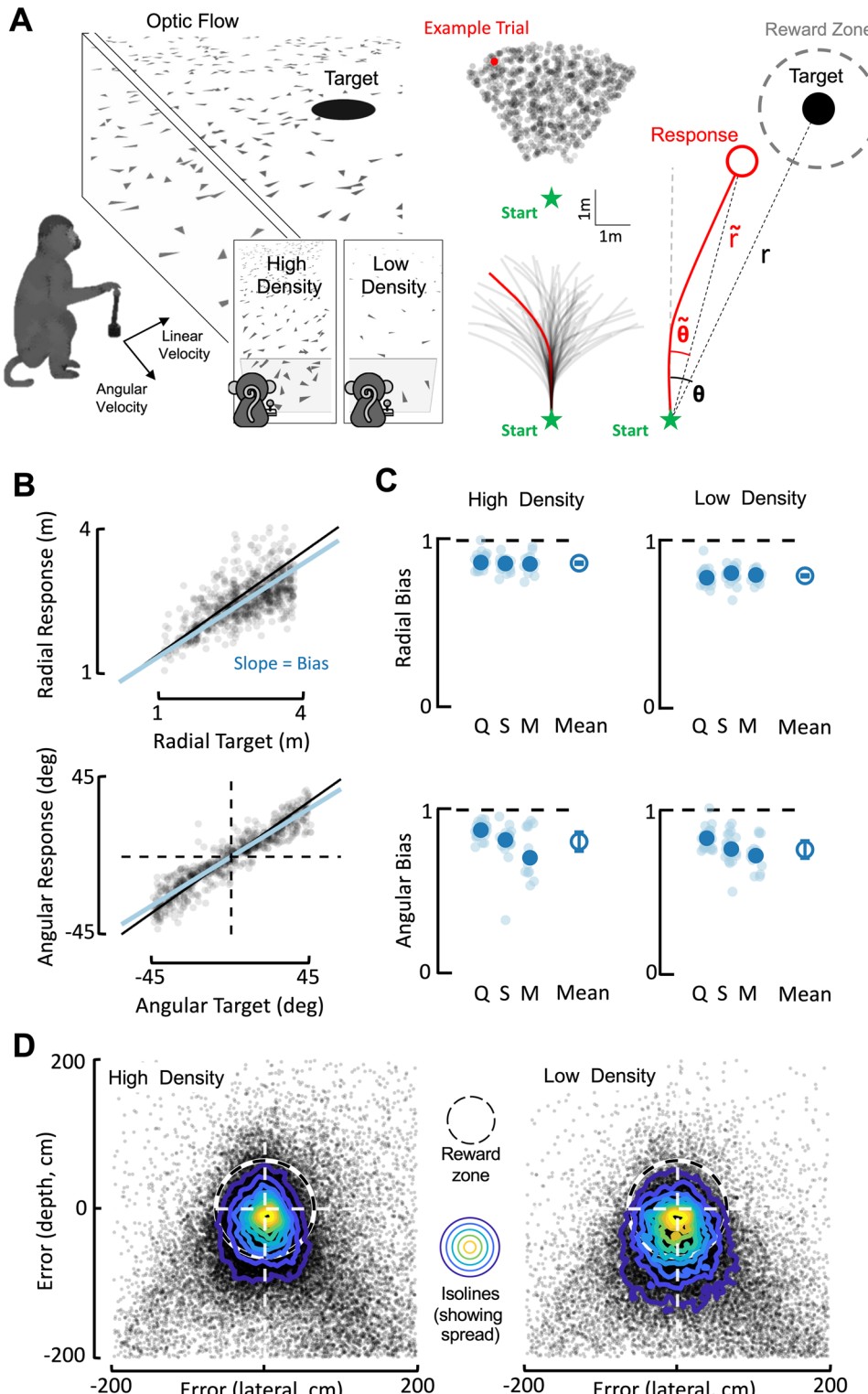

in the variance of trajectory endpoints is precisely what is expected when sensory likelihoods widen (i.e., decrease in sensory reliability) and priors are unbiased – as when animals were trained on the task. Instead, when observers are not trained on the task (feasible in humans but not in macaques), they have a biased slow-speed prior[27,37] and thus show both an increased variance[28] and overshooting[27] of targets during low-density optic flow path integration.

Eye movements were unconstrained during the task, which allowed animals to naturally saccade to target upon its presentation (Fig. S3A–D). Then, the animals pursued with their gaze the evolving location of the invisible target (Fig. S3A, E, F), which is seemingly an innate mnemonic task strategy[25,35] the animals employ. There was no difference in this eye-movement behavior (saccade or pursuit) as a function of optic-flow density (see Fig. S3 and its caption for statistical testing).

Together, these results demonstrate that macaques trained to navigate by integrating optic-flow velocity cues were able to generalize this behavior[26] and navigate by more heavily relying on an acquired internal model (mapping joystick position to virtual velocity) when sensory evidence was reduced.

**Fig. 1 | Macaques navigate in virtual reality to latent targets using path integration and an acquired internal model mapping joystick position to self-motion velocity. A** Setup. Monkeys use a joystick controlling their linear (max = 200 cm/s) and angular (max = 90 deg/s) velocity to navigate to the location of a briefly (300 ms) presented target. Monkeys were trained with a high-density of ground-plane elements, thus having access to a self-velocity signal (motion across their retina) they integrated into an evolving estimate of position. During recordings, we additionally used a low-density condition, wherein monkeys had to primarily rely on an internal model mapping joystick position to virtual velocity (akin to "stride size"). Targets were presented within a radial (1–4 m) and angular (−45 to 45 deg relative to straight ahead) range, and we quantified performance by regressing radial (r) and angular eccentricity (θ) of responses and targets. Figure adapted from REF[26] **B** Example session. Monkeys tended to undershoot targets in

radial distance (top) and eccentricity (bottom). For each session, we fit a linear regression (response = gain x target location + intercept) and thus a gain of '1' indicates no bias. Gains <1 indicate undershooting. **C** Radial and angular biases during high- and low-density conditions. Q, S, and M refer to three individual monkeys. Transparent small circles are individual sessions, and the larger opaque blue circle is the mean for each animal. The rightmost circle within each panel is the mean across animals, with error bars indicating ± 1 S.E.M. **D** Endpoint error and variance. Scatter plot of endpoint errors along the depth (y-axis) and lateral (x-axis) dimension as a function of density. For visualization, data is pooled across all sessions and monkeys (high-density $n$ = 44,476 trials; low-density $n$ = 43,286 trials). Isolines showing two dimensional histograms demonstrate the increased spread (variance) of endpoints relative to target in the low-density condition. Source data are provided as a Source data file.

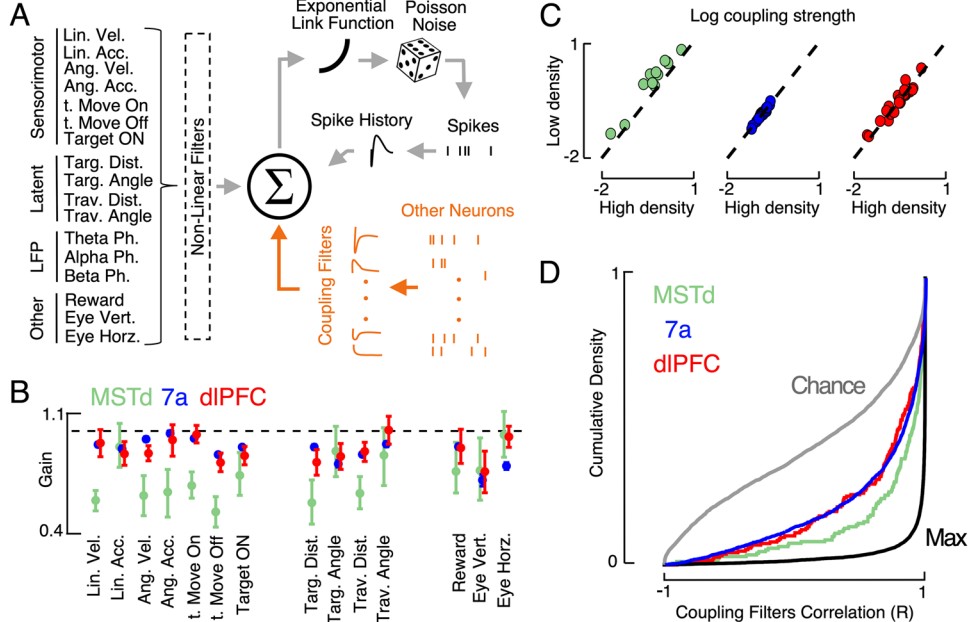

**Fig. 2 | Estimation of coupling filters and their remapping with optic flow density. A** P-GAM model. The encoding model (main text, see Fig. S9 for another variant) included 17 analog or digital task-variables, as well as spike history and coupling filters (in orange for emphasis). Figure adapted from REF[35] **B** Tuning function gain. Gain in tuning functions in the low-density condition relative to the high-density optic flow condition. A value under '1' (dashed black line) shows that the amplitude of the tuning function was reduced in the low-density condition relative to the high-density condition. Error bars are ± 95%CI. **C** Coupling filter strength. For all pairs of neurons that were coupled both in the high- and low-density conditions, we quantified the strength of their coupling and averaged within sessions. Coupling strength increased in the low-density condition in MSTd

(top; green) but was unchanged in area 7a (middle; blue) and dlPFC (bottom, red). Each dot is a session. **D** Coupling filter shape stability. Cumulative density functions (i.e., cumulative probability) of the coupling filters correlation coefficients (r) between low- and high-density conditions for all three brain regions (see Fig. S9 for the same data plotted as boxplot). Chance and ceiling-levels were determined by permutation, respectively correlating coupling filters recorded on separate sessions, channels, and animals (i.e., chance level), as well as correlating the same coupling pair (same session, animal, density, sender and receiver units) on odd and even trials (i.e., maximum level). MSTd (green) being the closest to "max" indicates that this region had the most stable (i.e., most correlated) coupling filters across density conditions. Source data are provided as a Source data file.

## Context differentially impacts neural statistical structure within sensory, parietal, and prefrontal cortices

We recorded single-unit spiking activity simultaneously in sensory (i.e., dorsomedial superior temporal area, MSTd, 240 neurons), parietal (area 7a, 2647 neurons) and prefrontal cortices (dorsolateral prefrontal cortex, dlPFC, 445 neurons; see Fig. S4 for images and MRI reconstruction of recording sites) to probe how cortical nodes, and in particular their statistical dependencies, support a population code and adaptive beliefs despite changing environments. Figure S5 shows peri-event time histogram (PETH) for example neurons in each brain area as a response to different events (i.e., target onset, movement onset, and offset) and across density conditions. To quantitatively summarize these responses, we fit a P-GAM[30] to account for stimulus-driven responses (e.g., linear and angular acceleration, velocity, and

position, distance to target and origin, vertical and horizontal eye position), as well as for elements of global neural dynamics (i.e., the ongoing phase of LFP within theta, alpha and beta ranges). The encoding models also included hundreds of potential coupling filters, capturing unit-to-unit time-resolved statistical regularities (exact number being session-specific; Fig. 2A, highlighted in orange). The P-GAM accounted well for spiking activity (average pseudo-$r^2$ = 0.091, ~1.5–2 times better than traditional Generalized Linear Models[38], see Fig. S6A for example tuning functions, both as raw binned responses and as estimated by the P-GAM), and did so better when including (vs. excluding) coupling filters (~1.5 times better in pseudo-$r^2$ and ~ 1.14 times in $R^2$, see Fig. S6B, C). Importantly, the addition of coupling filters did not change the estimate of tuning functions (correlations between tuning functions estimated with and without coupling functions,

$r \sim 0.93$, Fig. S6D), but critically, did allow to additionally account for signal-independent inter-neuron correlations (model prediction vs. empirical spike-triggered average, coupled model, $R^2 = 0.65 \pm 0.02$; uncoupled model, $R^2 = 0.26 \pm 0.06$; $t(3332) = 4.53$, $p = 0.0003$, Cohen's $d = 0.15$; see Fig. S6E for examples and caption for detail).

Regarding signal correlations, the fraction of neurons tuned to different task variables was not different across high- and low-density conditions (all $p > 0.25$, all Cohen's $d < 0.18$, Fig. S7A; see ref. 35 for a detailed characterization of signal correlations, including mixed selectivity[39-41], during the high-density condition). Further, provided that a neuron was tuned to a given task-variable in both high and low densities (96.8% of cases), this tuning was stable in shape across densities (examples in Fig. S7B, grand-average correlation coefficient = $0.92 \pm 0.007$, Fig. S7C). There was no difference in the stability of tuning function shape across densities in the different brain areas (one-way ANOVA, $F(2, 3196) = 0.61$, $p = 0.54$, $\eta^2 = 0.03$). Given this tuning stability, to quantify the impact of the different optic flow densities on the gain of neural tuning, we examined the gain of the linear regression between tuning functions in different densities of neurons stably tuned ($r > 0.5$, <95% of units; see Fig. S7B for example regressions). While we observed significant gain modulation in all brain regions (all $p < 0.005$), the effect size was considerable in MSTd (Cohen's $d = 0.21$) and negligible in the other areas (Cohen's $d < 0.06$ in 7a and dlPFC, Fig. 2B; examples in raw data in Fig. S5, and at the level of tuning functions in Fig. S7B). Neural responses were less prominent in MSTd (i.e., gains <1) during the low than the high-density condition, suggesting that while signal correlations were driven by changes in environmental input in MSTd, they were less so in 7a and dlPFC.

Regarding noise correlations, the fraction of units functionally coupled, either within (all $p > 0.88$) or across areas (all $p > 0.76$) was unchanged across the different environments (Fig. S8A, 92.1% of couples retained their coupling status across density conditions). On average, 20.4% of within-area coupling filters were significant, while only 1.08% of across area coupling filters were (11.31% when considering both within- and across-area pairs). Thus, for statistical power, all subsequent analyses regarding noise correlations focused on the within-area coupling filters.

The coupling strength (estimated as area under the coupling filter in kernel-space, collapsed across time) did not change with optic flow density within 7a or dlPFC (both $p > 0.67$, both Cohen's $d < 0.05$). On the other hand, coupling strength within MSTd increased during the low-density manipulation ($p < 0.005$, paired $t$-test, Cohen's $d = 0.51$; Fig. 2C). The evoked (i.e., peak response above or below baseline firing) spike-triggered average (in Hz-space, Fig. S6E) in coupled neurons in MSTd during the high-density condition was 0.64 Hz (± 0.09 Hz), and 1.02 Hz (± 0.11 Hz) in the low-density condition ($p = 0.011$). On average, inter-neuron spike-triggered averages were 1.13 Hz and 1.08 Hz, respectively in 7a and dlPFC (no difference across densities, both $p > 0.36$, both Cohen's $d < 0.12$). The correlation between change in tuning gain and coupling strength across sessions in MSTd showed a weak trend but was not significant ($r = -0.23$, $p = 0.19$). On the other hand, the correlation between tuning stability – as opposed to gain – showed a correlation with the number, or fraction, of neurons it coupled with (Fig. S8B), suggesting that unit-to-unit coupling stabilizes tuning responses across contexts.

To examine the finer grain temporal structure of neural co-fluctuations, we correlated coupling filters across density conditions (see Fig. S8C for example coupling filters in each area and across density conditions). We also correlated coupling filters estimated across different recording sessions (i.e., not simultaneously recorded) and as estimated on odd vs. even trials (within a single density), to respectively approximate chance and maximum correlation levels between coupling filters. This analysis showed that the temporal structure of stimuli-independent neural co-fluctuations partially remapped with context in all areas (greater than chance, smaller than max; Fig. 2D

shows full distributions as cumulative probabilities, see Fig. S9 for a different visualization), but did so less in MSTd (average $r$, $0.85 \pm 0.017$) than in 7a ($0.73 \pm 0.002$) and dlPFC ($0.73 \pm 0.014$, Kruskal-Wallis contrasting areas, $p = 0.0021$; post-hoc Holm-Sidak corrected contrasts, MSTd vs. 7a, $p = 0.003$; MSTd vs. dlPFC, $p = 0.02$; 7a vs. dlPFC, $p = 0.76$, Fig. 2D). To test for the generalizability of this finding, we fit an alternative encoding model wherein distances to target were expressed in egocentric coordinates (see Methods), and we included distances to the eventual monkey stopping location as opposed to distance from origin. Further, we included regressors accounting not only for eye position, but also for eye velocity (Fig. S9A–C). We confirm that also with an alternative encoding schema, areas 7a and dlPFC remapped noise correlations more readily than MSTd (Fig. S9D, Kruskal–Wallis, $H(2, 3332) = 12.82$, $p = 0.0023$, $\varepsilon^2 = 0.003$).

Together, these results suggest that while behaviorally the animals were able to maintain adaptive beliefs over their distance to target regardless of sensory environment (Fig. 1), the fine-grained statistical dependencies between neurons dynamically remapped with the changing computational demands (Fig. 2). The remapping was flexible in 7a and dlPFC (i.e., the actual shape of noise correlations changed), while in MSTd it was a strengthening of a fixed pattern of noise correlations (i.e., gain-modulation). Next, we questioned how a dynamically changing neural code may support a context-invariant belief supporting adaptive behavior.

## Stability of population code and relation to dynamic coupling filter remapping

We conjectured that the dynamic reconfiguration of fine-grained statistical dependencies between neurons may support flexible behavior by rendering task-relevant population dynamics invariant to the changing sensory input. And vice-versa, we hypothesized that without coupling filters remapping, distinct population dynamics would emerge under different contexts (due to the changing sensory input). Accordingly, we predict that 7a and dlPFC, but less so MSTd, would demonstrate context invariant population codes – despite (or even perhaps due to) their changing unit-to-unit couplings.

In the first step, we examined global population dynamics by principal component analysis (PCA). We binned trials according to target location (Fig. 3A, top row, overhead view) and optic-flow density (HD = "high-density", LD = "low density"). We then time-warped, averaged across trials, and visualized the first two PCs. The results show that while population dynamics changed with density in MSTd, they did not in 7a and dlPFC (Fig. 3A, second through fourth row. See Fig. S10 for quantification beyond the first 2 PCs and results showing that in higher dimensions the population code in dlPFC, but not 7a, does change with optic-flow density). The change in population dynamics in MSTd appeared to be a translation between optic-flow densities, as one would expect given the gain-modulation of tuning functions in this area (Fig. 2B). The lack of global change in population dynamics (as indexed by PCA) in 7a and dlPFC is also in line with the lack of single unit gain-modulation or remapping of signal correlations, the largest contributor to total neural variance.

Next, to quantitively test the hypothesis that the manifold coding for distance to target – as opposed to population dynamics as a whole (as in PC space - Figs. 3A and S10) – would remain stable in 7a and dlPFC, we used a linear population decoder (Lasso regression, 5-fold cross-validation) to estimate the animal's evolving distance to target. This distance to target is the key latent task variable leading to reward (rho <65 cm, see Methods) and accounts for evolving position relative to target along both azimuth and depth (distance to target = √ distance along azimuth$^2$ + distance along depth$^2$). Instead, animals can be rewarded for any angular distance to target (the target and reward zone being circles). Most importantly, to test for the stability of this task-relevant manifold, we assessed the ability of these decoders to

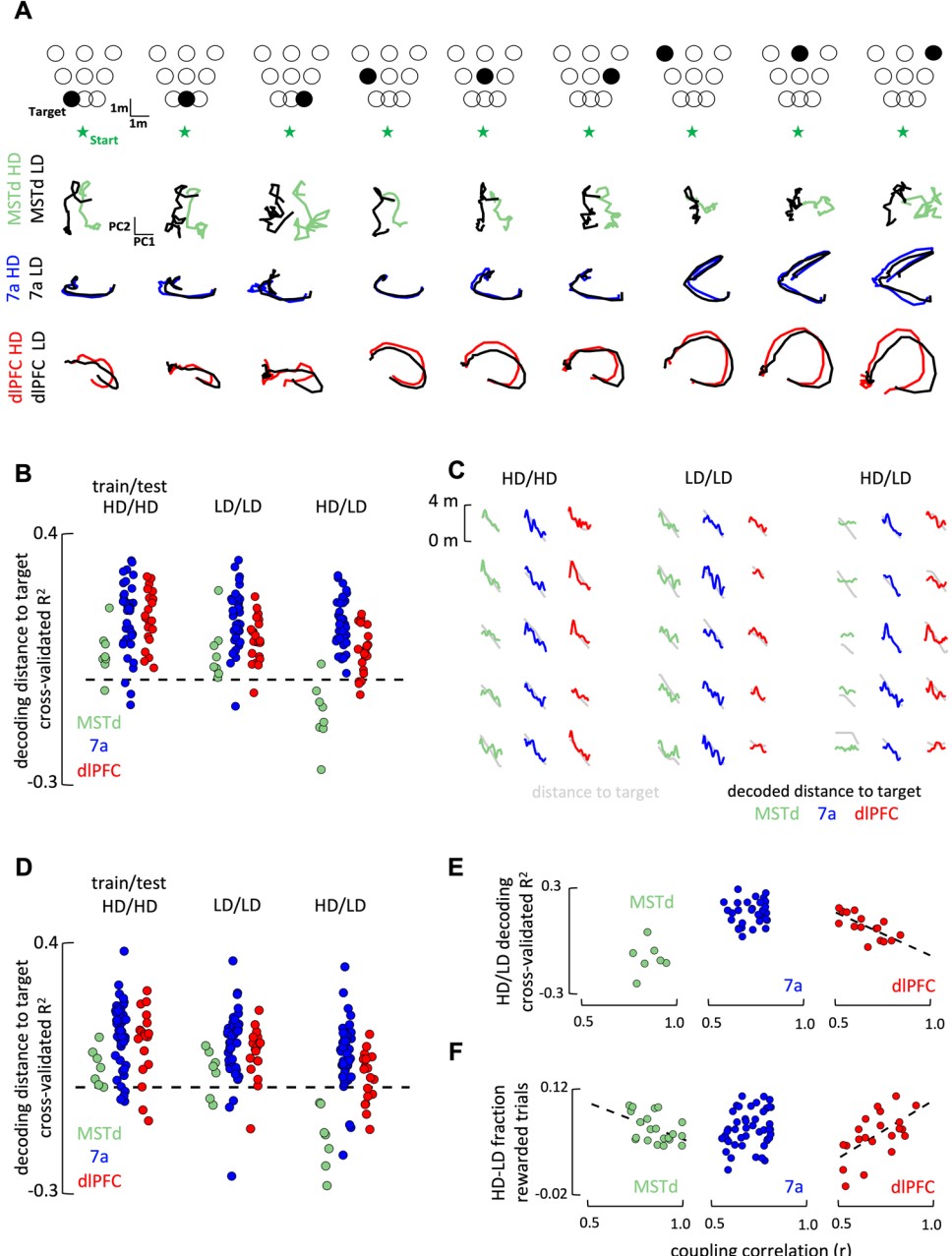

**Fig. 3 | Population codes and their relation to coupling filter remapping and behavior. A** Illustration of task-unaligned latent neural dynamics. Trials were categorized according to their target location (top: 9 bins, from near-to-far and from leftward-to-rightward) and density condition (bird's eye view, first row). Second, third, and fourth row respectively show the averaged latent trajectory in 2D for MSTd (green), 7a (blue), and dlPFC (red), both during high (colored) and low (black) density conditions, and according to target location. **B** Decoding of the distance to target based on neural population dynamics within condition (high-density/high-density, low-density/low-density) or across (high-density/low-density). Dots are individual sessions, dashed black line is cross-validated $R^2 = 0$. **C** Five example trials decoding distance to target. **D** Decoding of the distance to target based on populations where signal correlations were left intact but noise correlations were destroyed. **E** Relation between stability of the fine-grain neural co-fluctuations and ability of the decoder to generalize. **F** Relation between stability of the fine-grain neural co-fluctuations and session-by-session changes in behavioral performance with optic flow density. Source data are provided as a Source data file.

generalize across density conditions (i.e., train on high-density and test on low-density). Results showed that while all areas were able to decode distance to target when trained and tested within a single context (Fig. 3B, left and center, cross-validated $R^2 > 0$, all $p < 0.029$, all Cohen's $d > 0.31$), the decoders generalized across contexts in area 7a and dlPFC (CV $R^2 > 0$, respectively, $t(32) > 1.5 \times 10^7$, $p = 4.19 \times 10^{-15}$, Cohen's $d > 3$ and $t(19) = 5.13$, $p = 2.57 \times 10^{-5}$, Cohen's $d = 1.14$), but not in MSTd (CV $R^2 < 0$, $t(7) = 3.49$, $p = 0.01$, Cohen's $d = 1.23$, Fig. 3B, rightmost, see Fig. 3C for example single trial decoding of distance to

target). This suggests the existence of a similar population readout for distance to target across contexts in 7a and dlPFC, but not MSTd.

Three pieces of evidence suggest that the stability of the manifold coding for distance to target in dlPFC, but not that in 7a, is related to the dynamic remapping of temporally-resolved unit-to-unit couplings across densities.

First, artificially eliminating the statistical dependencies between neurons (see Methods and Fig. S11) abolished the ability of dlPFC (CV $R^2 > 0$, $t(19) = 0.21$, $p = 0.84$, Cohen's $d = 0.047$), but not area 7a (CV

$R^2 > 0$, $t(32) = 6.53$, $p = 1.17 \times 10^{-8}$, Cohen's $d = 1.13$), in decoding distance to target across densities (HD/LD decoding, Fig. 3D). Eliminating these neural co-fluctuations did not change global neural dynamics as quantified by the first two PCs, which were primarily driven by signal and not noise correlations. Further, this effect was specific to the coding of distance to target: given that a brain area was able to decode a certain task variable (e.g., 7a decoding linear velocity), eliminating the corresponding noise correlations did not abolish this area's ability for within- or across-context decoding (Fig. S12A). Moreover, the impact of noise correlations on the ability to cross-decode distance to target in dlPFC was not specific to linear decoders, but also extended to a non-linear artificial neural network (Fig. S12B).

Second, there were strong and specific session-to-session correlations between the stability of coupling filters in dlPFC and the ability to decode the key task variable across contexts. That is, the more unit-to-unit couplings remapped within dlPFC in a session, the better was this session's ability to decode distance to target across density contexts ($r = -0.71$, $p = 0.0018$; Fig. 3E). This was not true in dlPFC for any other task variable (Fig. S13), nor was it for area 7a, where coupling filter stability did not correlate with decoding generalizability for distance to target ($r = 0.08$, $p = 0.65$). In fact, where we did observe a correlation between the ability for cross-context decoding and stability of coupling functions in 7a was for linear acceleration ($r = 0.59$, $p = 7.1 \times 10^{-5}$, Fig. S13). The stability of tuning functions across densities did not correlate with decoding generalizability of distance to target in 7a ($r = 0.18$, $p = 0.21$) or dlPFC ($r = 0.23$, $p = 0.12$).

Third, the degree of remapping of neural co-fluctuations in dlPFC (and its stability in MSTd) were related to behavioral performance. Indeed, we observed that the more coupling filters remapped in dlPFC ($r = 0.59$, $p = 0.004$), the less was behavior impacted by the loss of sensory evidence in the low-density condition (Fig. 3F). Interestingly, we observed the opposite effect in MSTd, with more stable coupling filters resulting in behavioral performance being less impacted by the loss of sensory evidence ($r = -0.47$, $p = 0.030$). There was no relation between behavioral performance and coupling filter stability in area 7a ($r = 0.18$, $p = 0.22$). Lastly, demonstrating the specificity of these effects, there was no relation between decoder generalizability or behavioral performance and changes in coupling strength (all $|r| < 0.09$, $p > 0.58$), as opposed to shape.

Together, these results suggest a context-invariant population code – both global and aligned to the critical task variable, distance to target – in area 7a and dlPFC. These areas are also those demonstrating significant remapping of unit-to-unit couplings with context (Fig. 2D). Correlational analyses and controls artificially eliminating noise correlations suggest a relation between the stability of the task-relevant population code and coupling filters remapping in dlPFC but not in 7a.

## Discussion

We leveraged a task allowing for rapid generalization[26] and a naturalistic interplay between sensory input, internal neural dynamics, and motor output in an attempt to bridge the gap between functional connectivity, population codes, and flexible behavior. To do so, we also leveraged a statistical methods innovation[30] to estimate time-resolved coupling filters and their statistical contribution to a neuron's spiking activity. This approach opens a window of opportunity above and beyond the traditional estimates of noise correlation[29,34], which are not time-resolved and cannot be easily applied to complex naturalistic tasks.

The most remarkable observation is that co-fluctuation of neural activity changed differently within sensory and parietal/frontal cortices when animals were tasked to further rely on acquired internal models, as opposed to sensory evidence. While MSTd seemingly attempts to rigidly extract environmental signals (i.e., increased strength in noise correlations albeit with fixed dynamics), the fine-grain statistical dependencies between neurons in 7a and dlPFC

flexibly reconfigured under different contexts. This observation is reminiscent of a wealth of human neuroimaging studies[2–5] emphasizing the flexible remapping of the fronto-parietal network with changing task demands. Our results also concord with previous recordings in macaques showing context or task dependent modulations in spiking activity in prefrontal[6–8] and parietal[42] cortices, but not extrastriate[42] cortex. However, at difference from those studies, we must note that here we do not observe context-dependent differences in average firing rates (i.e., tuning functions) in 7a and dlPFC, but in the pairwise interactions of neurons within these areas. To the best of our knowledge, prior primate studies have not assessed if and how time-resolved coupling filters change with context across cortical areas.

The relation between remapping of coupling filters, population codes, and flexible behavior also differed between dlPFC and 7a. Seemingly, dlPFC flexibly reconfigured to maintain an adaptive latent representation of the key task variable (i.e., "computation through dynamics"[43]). This is supported by the fact that the degree to which dlPFC remapped across contexts in different sessions related to the stability of the distance to target read-out manifold, as well as to behavioral performance. Further, eliminating the statistical dependencies between neural responses in dlPFC abolished the ability of this area for cross-context (linear or non-linear) decoding of the central task-relevant variable: distance to target. Area 7a similarly exhibited a context-invariant population code, yet this ability appeared unrelated to the remapping of coupling filters. This latter observation suggests that the context-invariant population dynamics in area 7a are potentially downstream (i.e., inherited) from those in dlPFC, that this area is less driven by visual flow (or the lack thereof) than it is by body-related (e.g., vestibular, proprioceptive, motor) or decisional signals (see refs. 35,44,45 for evidence to this regard), and/or that relative to dlPFC, population responses in 7a were less driven by noise correlations and more by signal correlations and/or local field potentials. Fittingly, we observe that higher PCA dimensions (i.e., accounting for less variance and thus less driven by task inputs and more by noise correlations) varied with context in dlPFC but not in 7a (Fig. S10), and that coupling stability related to the ability to cross-decode linear acceleration (Fig. S10). Similarly, in prior work[35] we have demonstrated that area 7a is strongly driven by sensorimotor variables (e.g., self-motion velocity, which is driven by joystick/hand position) and the ongoing phase of LFP in different frequency bands. Thus, while the population code in area 7a was context-invariant, this appears to be less dependent on the structure of unit-to-unit statistical co-fluctuations.

Mechanistically, some[6] have argued that a change in context results in an alteration in recurrent dynamics. Others[7] have assumed that connectivity does not change at this time scale, and thus neither can the underlying neural dynamics. Instead, according to this latter view, changes in context ought to alter the initial condition or input into the dynamical system (also see ref. 46). At first glance, it would appear that our results support the former interpretation, given that we report changes in functional connectivity. However, we must highlight that given the closed-loop nature of our task[25–28,47], input to the system is state-dependent and continuously changing. This renders challenging, if not impossible, the teasing apart of the relative contributions of internally generated dynamics from external inputs[48]. Future work may be able to resolve these questions, by contrasting biological data to RNNs trained to perform the same task. In fact, theoretical work suggests that flexible and cross-context computations rely on the re-purposing of modular dynamical motifs[49], and our group has recently demonstrated that modular RNNs more readily allow agents to generalize behavior within this naturalistic, navigate-to-target task[22]. At first approximation, here we may speculate that in the macaque the algorithmic modularity[22,46,49] needed for flexible computations in closed-loop behaviors resides in dlPFC, with MSTd more rigidly reflecting the sensory environment and 7a more closely associated with the motor aspects of the task and perhaps "reading" from dlPFC.

More importantly, in real-world behaviors, our brains, bodies, and environments may be best considered a single dynamical system, so that teasing apart the role of internal neural dynamics and inputs from the environment may be a false dichotomy. Instead, we highlight that when faced with different environments posing different computational challenges, the brain appears to adopt a two-pronged approach. Sensory areas may not change their apparent functional connectivity, while frontal areas dynamically remap. We propose that this approach may allow for a compromise between faithfully reflecting our sensory environment (i.e., representation in sensory areas) and not completely changing our interpretation of the world around us (i.e., coding of latent variables or beliefs in prefrontal cortices) given stochastic fluctuations in environmental input.

## Methods

### Animals and animal preparation

Three adult male rhesus macaques (*Macaca Mulatta*; 9.8-13.1 kg, all male, 7–8 years old) were studied. We analyzed behavioral and neural data from 27 sessions in monkey Q, 38 sessions in monkey S, and 17 sessions in monkey M (82 sessions in total). Experimental data was collected by researchers blind as to the rationale and hypothesis of the experiment. All animals performed the same task, where trial-type was manipulated on a trial-by-trial basis, and thus there was no allocation of animals into experimental groups. These same sessions have been reported on before[35], but only the "high-density" trials. The "low-density" data and their comparison to the "high-density" condition was not presented previously. All animals were chronically implanted with a lightweight polyacetal ring for head restraint, and scleral coils for monitoring eye movements (CNC Engineering, Seattle WA, USA). Further, for acute recordings, animals were outfitted with a removable grid to guide electrode penetrations. Prior to the surgery, the brain areas of interest were identified using structural MRI to guide the location craniotomies. Surgeries followed standard aseptic techniques and the health and welfare of animals was continuously monitored by laboratory and veterinary staff post-operatively by checking responsiveness, appetite, vocalizations, posture, mobility, pain, hydration, feces, urine, and potential swellings and discharges. Monkeys were trained via standard operant conditioning to path integrate to the location of briefly visible targets ("fireflies") by accumulating optic flow velocity signals caused by active self-motion and thus motion of optic flow elements across their retina. All surgeries and procedures were approved by the Institutional Animal Care and Use Committee at Baylor College of Medicine and New York University, and were in accordance with National Institutes of Health guidelines.

### Experimental setup

Monkeys were head-fixed and secured in a primate chair. A 3-chip DLP projector (Christie Digital Mirage 2000, Cypress, CA, USA) rear-projected images onto a 60 × 60 cm screen that was attached to the front of the field coil frame, 32.5 cm in front of the monkey. The projector rendered stereoscopic images generated by an OpenGL accelerator board (Nvidia Quadro FX 3000 G). To navigate, the animals used an analog joystick (M20U9T-N82, CTI electronics) with two degrees of freedom to control their linear (max = 200 cm/s) and angular (max = 90 deg/s) speeds in a virtual environment. This virtual world comprised a ground plane whose textural elements had limited lifetime (250 ms). The ground plane was circular with a radius of 70 m (near and far clipping planes at 5 cm and 40 m respectively), with the subject positioned at its center at the beginning of each trial. Each texture element was an isosceles triangle (base x height: 8.5 × 18.5 cm²) that was randomly repositioned and reoriented at the end of its lifetime. The standard density of the ground plane elements (i.e., the density on which animals were trained) was 5.0 elements/m² (high-density condition). This density was reduced by a factor of 50 (0.1 elements/m²) in a manipulation condition to force the animals to further use their acquired internal

model, rather than solely sensory evidence. All stimuli were generated and rendered using C ++ Open Graphics Library (OpenGL) by continuously repositioning the camera based on joystick inputs to update the visual scene at 60 Hz. The camera was positioned at a height of 0.1 m above the ground plane. Spike2 software (Cambridge Electronic Design Ltd., Cambridge, UK) was used to record and store the timeseries of target and animal's location, animal linear and angular velocity, as well as eye positions. All behavioral data were recorded along with the neural event markers at a sampling rate of 833.33 Hz.

### Behavioral task

Monkeys steered to a target location (circular disc of radius 20 cm) that was cued briefly (300 ms) at the beginning of each trial. Each trial was programmed to start after a variable random delay (truncated exponential distribution, range: 0.2–2.0 s; mean: 0.5 s) following the end of the previous trial. The targets appeared at a random location between −45 to 45 deg of visual angle, and between 1 and 4 m relative to where the subject was stationed at the beginning of the trial. The joystick was always active, and thus monkeys were free to start moving before the target vanished, or before it appeared. Monkeys typically performed blocks of 500-750 trials before being given a short break. In a session, monkeys would perform 2 or 3 blocks. High (5.0 elements/m²) and low (0.1 elements/m²) density conditions were randomly intermixed with equal probability within each block. On average, animals attempted 1070.26 trials per session, resulting in a total dataset of 44,476 trials in the high-density condition and 43286 in the low-density condition. Feedback, in the form of juice reward was given following a variable waiting period after stopping (truncated exponential distribution, range: 0.1–0.6 s; mean: 0.25 s). They received a drop of juice if their stopping position was within 0.6 m away from the center of the target ("reward boundary"). No juice was provided otherwise. In order words, animals were rewarded if they stopped within a radial distance of 0.6 m from the center of the target. Angular distance (range = 0–360°) did not impact the delivery of reward – i.e., animals could over- or under-shoot targets and still be rewarded. Monkeys were trained by gradually reducing the size of the reward zone until their performance stopped improving.

### Neural recording and pre-processing

We recorded extracellularly, either acutely using a 24-channel linear electrode array (100 μm spacing between electrodes; U-Probe, Plexon Inc, Dallas, TX, USA; MSTd and 7a in Monkeys Q and S, dlPFC in monkey M) or chronically with multi-electrode arrays (Blackrock Microsystems, Salt Lake City, UT, USA; 96 electrodes in area 7a in Monkey Q, and 48 electrodes in both area 7a and dlPFC in monkey S). Chronic setups were used for their higher yield, while acute linear probe recordings were used in order to target MSTd, which is deep. Further, we used both chronic and acute preparations to ascertain that neural results were independent of the type of probe used (Fig. S14). During acute recordings with the linear arrays, the probes were advanced into the cortex through a guide-tube using a hydraulic microdrive. Spike detection thresholds were manually adjusted separately for each channel to facilitate real-time monitoring of action potential waveforms. Recordings began once waveforms were stable. The broadband signals were amplified and digitized at 20 KHz using a multichannel data acquisition system (Plexon Inc, Dallas, TX, USA) and were stored along with the action potential waveforms for offline analysis. Additionally, for each channel, we also stored low-pass filtered (-3dB at 250 Hz) local-field potential (LFP) signals. For the array recordings, broadband neural signals were amplified and digitized at 30 KHz using a digital headstage (Cereplex E, Blackrock Microsystems, Salt Lake City, UT, USA), processed using the data acquisition system (Cereplex Direct, Blackrock Microsystems) and stored for offline analysis. Additionally, for each channel, we also stored low-pass filtered (-6 dB at 250 Hz) local-field potential (LFP) signals sampled at 500 Hz.

Finally, copies of event markers were received online from the stimulus acquisition software (Spike2) and saved alongside the neural data.

Spike detection and sorting were initially performed on the raw (broadband) neural signals using KiloSort 2.0 software on an NVIDIA Quadro P5000 GPU. The software uses a template matching algorithm both for detection and clustering of spike waveforms. The spike clusters produced by KiloSort were visualized in Phy2 and manually refined by a human observer, by either accepting or rejecting KiloSort's label for each unit. In addition, we computed three isolation quality metrics; inter-spike interval violations (ISIv), waveform contamination rate (cR), and presence rate (PR). ISIv is the fraction of spikes that occur within 1 ms of the previous spike. cR is the proportion of spikes inside a high-dimensional cluster boundary (by waveform) that are not from the cluster (false positive rate) when setting the cluster boundary at a Mahalanobis distance such that there are equal false positives and false negatives. PR is 1 minus the fraction of 1 min bins in which there is no spike. We set the following thresholds in qualifying a unit as a single unit: ISIv <20%, cR <0.02, and PR > 90%. After spike sorting, the final neural dataset consisted of 3332 single units (MSTd: 240; parietal area 7a: 2647; dlPFC, 445 neurons), with an average of 40 units simultaneously recorded per session. The average firing rate of units in MSTd, 7a, and dlPFC were, respectively, 5.69, 6.14, and 6.61 spikes/s.

## Analyses

**Behavioral Analyses.** The location of targets and the monkey's end locations were expressed in polar coordinates, with a radial distance (target = r, response = r~ ) and eccentricity from straight ahead (target = θ; response = θ~ ). On a subset of trials (-13%) animals stopped after <0.5 m, suggesting they aborted the trial. Similarly, on a subset of trials (-5%) animals did not stop during the course of a trial (max duration = 7 seconds). These trials were discarded before further behavioral analyses. A linear model with multiplicative gain (i.e., response = gain x target location + intercept; see refs. 25–28,37 for the same approach) accounted well for the observed data (average $R^2 = 0.72$). Thus, we used the gain term of the corresponding linear regressions as a measure of behavioral performance (gain = 1 would indicate no over- or under-shooting of targets, gains <1 indicate undershooting, and gains > 1 indicate overshooting. The average intercept = 1.1 cm in radial distance and 0.2 deg, neither significantly different from 0, both $p > 0.81$; Fig. 1B, C). We also report the fraction of trials in which the animal was rewarded (reward radius = 65 cm centered on the firefly, determined by staircase such that animals would be rewarded on -66% of trials). Lastly, to index variance in endpoint responses we compute for each session the S.E.M of distances to target (rendering a two-dimension error, along x and y, into a single dimension, distance). We then report the mean S.E.M and variance across sessions and density conditions.

**Generalized Additive Model.** The Poisson Generalized Additive Model (P-GAM) defines a non-linear mapping between spike counts of a unit $y_t \in \mathbb{N}_0$ (binned at 6 ms temporal resolution) and a set of continuous covariates $\mathbf{x}_t$ (angular and linear velocity and acceleration, angular and linear distance traveled, angular and linear distance to target, and LFP instantaneous phase across different frequency ranges), and discrete events $\mathbf{z}_t$ (time of movement onset/offset, target onset, reward delivery, and the spike counts from simultaneously recorded units). The resultant tuning functions for continuous covariates ($\mathbf{x}_t$) will have as x-axis values along the range of the particular variable (e.g., cm, cm/s), while the tuning functions for discrete events ($\mathbf{z}_t$; time of target onset, movement onset and offset, and time of reward) will have time (e.g., seconds) as x-axis. A unit's log-firing rate was modeled as a linear combination of arbitrary non-linear functions of the covariates,

$$\log \mu = \sum_j f_j(x_j) + \sum_k f_k * z_k \tag{1}$$

where * is the convolution operator, and the unit spike counts are generated as Poisson random variables with rate specified by (Eq. 1). Input specific non-linearities f(·) are expressed in terms of flexible B-splines, $f(\cdot) \approx \boldsymbol{\beta} \cdot \mathbf{b}(\cdot)$ and are associated with a smoothness enforcing penalization term controlled by a scale parameter $\lambda_f$,

$$PEN(f, \lambda_f) = -\frac{1}{2}\lambda_f \boldsymbol{\beta}^T S_f \boldsymbol{\beta}, S_f = \int b'' \cdot b''^T dx \tag{2}$$

The larger $\lambda_f$, the smoother the model. This penalization terms can be interpreted as (improper) Gaussian priors over model parameters, the resulting log-likelihood of the model takes the form,

$$\mathscr{L}(\mathbf{y}) = \log p(\mathbf{y}, |, \mathbf{x}, \mathbf{z}, \boldsymbol{\beta}) + \sum_f PEN(f, \lambda_f) \tag{3}$$

with $\mathbf{y} \in \mathbb{R}^T$ the spike counts of the unit, $\mathbf{x} \in \mathbb{R}^{J \times T}$ and $\mathbf{z} \in \mathbb{R}^{K \times T}$, T the time points, $\boldsymbol{\beta}$ the collection of all B-spline coefficients and p(·) the Poisson likelihood. Both parameters $\boldsymbol{\beta}$ and the hyperparameters $\lambda$ are learned from the data by an iterative optimization procedure that switches between maximizing Eq. 3 as a function of the parameters, and minimizing a cross-validation score as a function of the hyperparameters (see ref. 30 for further details and see refs. 30,35 for prior extensive validation of the model). The probabilistic interpretation of the penalization terms innovatively allowed us to compute a posterior distribution for the model parameters, derive confidence intervals with desirable frequentist coverage properties, and implement a statistical test for input variable inclusion that selects a minimal subset of variables explaining most of the variance.

In addition to the main encoding model described above and in the main text, we also fit an alternative encoding model to test for generalizability of the reported effects – particularly those relating to the remapping of functional connectivity with optic-flow context in 7a and dlPFC. This alternative encoding model is defined by a different set of inputs. Namely, we include the distance to target in egocentric coordinates as opposed to a virtual three-dimensional space. Namely, we project the target onto the two dimensions of the screen, and then compute the vertical and horizontal angle between the current position of gaze, the eyes, and where the target is on screen-coordinates. We also include (in egocentric coordinates) the distances to the eventual stopping position, given that animals are tasked with stopping where they believe the target to be positioned. And thus the stopping location is a proxy for believed target position[27]. Lastly, given that pursuit signals in MT/MST/MSTd have non-linear dynamics[50-52], we augment the model to include eye velocity components (horizontal and vertical) in addition to eye position. Differences between the encoding model used in the main text and the alternative model are illustrated in Fig. S9A and B.

**Pseudo-$R^2$.** We computed pseudo-$R^2$s to assess fit quality. This metric is a goodness of fit measure that is suitable for models with Poisson observation noise[53], and is computed as,

$$pseudoR^2 = 1 - \frac{\mathscr{L}(\mathbf{y}) - \mathscr{L}(\hat{\mathbf{y}})}{\mathscr{L}(\mathbf{y}) - \mathscr{L}(\bar{\mathbf{y}})} \tag{4}$$

with $\mathscr{L}(\mathbf{y})$ being the likelihood of the true spike counts, $\mathscr{L}(\hat{\mathbf{y}})$ being the likelihood of the P-GAM model prediction, and $\mathscr{L}(\bar{\mathbf{y}})$ being the likelihood of a Poisson null-model (constant mean rate). We computed this measure on held-out test trials (20% of the total trials, not used for inferring model parameters). The pseudo-$R^2$ can be interpreted as the fraction of the maximum possible likelihood (normalized by the null model) that the fit attains. The score is 0 when the P-GAM fits are no more likely than the null model (constant mean rate), 1 when it perfectly matches the data, and can be negative when overfitting occurs. In this latter case, we excluded the unit from analysis (2.7% of the recorded units). Empirically, the pseudo-$R^2$ is a stringent metric[54]

and ranges in values that are substantially lower than the standard $R^2$ (see Fig. S6B for a comparison of pseudo-$R^2$ in coupled and uncoupled P-GAMS). We also provide an $R^2$ metric (average = 0.26, no different across densities, $p = 0.91$) as a comparison to prior studies (current study is better than its most similar report[44] with a reported average of 0.15) and for ease of interpretability, but it must be noted that this latter metric relies on first convolving spike trains (here we used a Gaussian filter with standard deviation = 20 ms, Fig. S6C) and will change depending on the convolution filters used.

Coupling Filters. Coupling filters (and the corresponding inferential statistics) were determined via the P-GAM[30]. Within area coupling filters were set to a duration of 36 ms, and across area filters were initially set to a duration of 600 ms. We focus our analyses in the main text exclusively on the within area coupling filters, given their higher presence rate, and the fact that we can be more confident in indexing unit-to-unit interactions when restricting the analyses to a short timeframe (i.e., 36 ms). For further scrutiny of the coupling filters and their contrast to empirically-derived cross-neuron spike-triggered averages (Fig. S6E) we additionally fit encoding models with longer coupling (1.2 seconds), or without coupling filters altogether (e.g., Fig. S6D, E). For the coupling probabilities reported in Fig. S8A, we corrected for the effects of unit distance (i.e., Fig. S1A) by first fitting a brain region-specific logistic regression. Specifically, we expressed coupling probability as a non-linear function of electrode distance as follows,

$$p(c = 1) = \text{logit}^{-1}(f(d)) \tag{5}$$

With c being a binary variable taking value 1 for significant coupling and 0 otherwise, d being the electrode distance, and f being a nonlinear function expressed in terms of B-splines. Each brain area was fit independently, and the coupling probability in Fig. S8A was set as the model prediction for a distance of 500um.

Coupling probability was computed as the fraction of significantly coupled units within area pairs (p-value < 0.001 covariate inclusion test). Coupling strength (i.e., Fig. 2C) was computed as the norm of the betas defining the coupling filter. We also report evoked spike-triggered averages (i.e., compute PETH of "receiver" unit at spike onset of "sender" units) of units the P-GAM deemed to be functionally coupled, as a function of brain area and optic-flow density. Coupling stability (e.g., Fig. 2D) was computed as the correlation between coupling filters in the low- and high-density condition. Chance and ceiling levels were established by permutation. To relate the change of within area functional connectivity with the change in behavior caused by the decrease in optic flow signals, we computed the average coupling stability per area and session, given the session had at least 5 coupled units and a P-GAM pseudo-$r^2$ greater than 0.01.

Latent Population Dynamics. We computed population latent dynamics by principal component analysis (PCA). For the illustration in Fig. 3A, PCA was based on trial averaged firing rates in rewarded trials across all sessions (i.e., 'pseudo-populations). We segmented trials between the time of target onset and movement offset, and estimated instantaneous firing rates by convolving spike counts (binned at 6 ms) with a 100 ms Gaussian window. We matched trial durations by linearly interpolating the rates (i.e., time-warping). Then, for the visualization, we grouped trials into eighteen categories (9 target positions x 2 densities; see Fig. 3A, top) and averaged the instantaneous rate within each category. We then stacked all rates (for all time points and categories) in a single vector $\mathbf{r}_i \in \mathbb{R}^{T \cdot C \cdot D}$, where i = 1, . . . ,N is the neuron index, T is the number of the interpolating time points, C is the target location (categorized), and D are the density conditions. We computed the PCA projection based on the $(T \cdot C \cdot D) \times N$ matrix of the population rates, and finally projected each condition separately on the PCA axis. To quantitatively assess the stability of global population dynamics across density conditions we developed a permutation test that compares the dynamics of similar trials within and between density conditions. For each session we first extracted the PCA axis by stacking the instantaneous firing rate of all rewarded trials (estimated as above). Subsequently, we applied the following algorithm 1000 times per session: 1) randomly select a reference trial from the high-density condition; 2) selected two trials for comparison, a high-density trial and a low-density trial, for which the initial target location was less then 10 cm away from that of the reference trial; 3) project the population activity of these 3 trials onto the PCA axis; 4) compute the Frobenius norm-based distance between the projected activity of the target trial and each of the two comparison trials. This procedure yielded the distributions shown in Fig. S10A (5 example sessions shown). Finally, we computed the d-prime score separating these distributions (Fig. S10B). This procedure was repeated while projecting neural activity on 1 to 10 PCs. Similarly, we repeated this analysis while conducting PCA separately on the different density conditions, again yielding conceptually identical results.

Decoding, Lasso Regression. To specifically examine the manifold coding for distance to target, we decoded this variable (the critical task-relevant latent variable, given that the target is invisible) from the population of instantaneous firing rates estimated by convolving spike counts with a 100 ms Gaussian window. For a fair comparison of the decoding accuracy between areas, we matched the number of units (10 units) and the mean firing rate for each area on a session-by-session basis. Then, we split trials according to the density condition and, within density, we used 90% of trials for training a linear decoder. The remaining 10% of data was used to estimate the decoder prediction accuracy. For cross-context decoding we trained on high-density optic flow and decoded on the low-density condition. We opted for a Lasso regularized regression to control for overfitting. Lasso hyperparameters were selected by grid-search with a 5-fold cross-validation scheme. Model prediction accuracy within and across density condition was quantified as cross-validated $r^2$ (Fig. 3B, example distances to target and their reconstruction shown in Fig. 3C). To test for the role of neural co-fluctuations in allowing for distance to target decoding, we adopted a shuffling procedure that destroyed neuron-to-neuron correlations but kept signal correlations intact by matching the variable of interest in shuffled and unshuffled data (see Fig. S11). Namely, we first discretized distance to target in 15 bins (other discretization tested and yielded conceptually identical results). Then, since each time point corresponded to a particular bin, for each neuron we shuffled the counts for each time point corresponding to the same bin. This procedure leaves task variables and tuning functions unchanged (as we are shuffling like-for-like) but disrupts the correlation between neurons. We then conducted the lasso regression as detailed above. This shuffling procedure was done 10 times per neuron and session to assure reproducibility (Fig. 3D plots the first run, after checking results were reproduced across all shuffling runs). Further, we implemented the same analysis when matching not only the variable we were interested in decoding (i.e., distance to target), but also a second variable: linear velocity (Fig. S11). This further accentuates the correlation between task-variables (and thus signal correlations) in shuffled (excluding noise correlations) and unshuffled (including noise correlations) datasets (Fig. S11C). The main result with cross-context decoding of distance to target being impacted by the lack of noise correlations in dlPFC but not in 7a was replicated (Fig. S11D). Lastly, to test for specificity, we perform the same analysis on all task variables (Fig. S12, matching signal correlations for the variable we are decoding), and while using either linear (Fig. S12A) or non-linear decoders (Fig. S12B)

Decoding, Non-linear Artificial Neural Network (ANN). To test whether the reported impact of noise correlations on the ability of dlPFC to cross-decode the key latent task-variable is specific or not to linear decoding, we build a 2-layer ANN to non-linearly decode continuous task variables (e.g., distance to target, linear velocity, etc.). Data pre-processing was akin to that for the Lasso regression (above), and we used sessions with at least 10 units simultaneously recorded in a given

area. The first layer of the ANN (input) had 10 neurons, with the second layer being an expansive layer with 20 neurons. There was an output layer of a single neurons. The input and hidden layers had ReLu activation functions, and we used a Mean Squared Error (MSE) loss function with L2 regularization to prevent overfitting. The training was performed using the Adam optimizer from the Optax library[55] with adaptive learning rates. Weights were initialized using a normal distribution and with no bias. Overall, the non-linear decoder performed better than the Lasso regression (mean CV $R^2$ across all variables, within context: regression = $0.171 \pm 0.012$; ANN = $0.196 \pm 0.015$, $p = 0.019$; across-context: regression = $0.155 \pm 0.015$; ANN = $0.169 \pm 0.018$, $p = 0.023$). The specific loss of the ability to decode distance to target across optic-flow contexts in dlPFC remained (Fig. S12B).

### Reporting summary
Further information on research design is available in the Nature Portfolio Reporting Summary linked to this article.

### Data availability
Source data are provided with this paper. The data used in this study are available in the OSF database: https://osf.io/eybc8. Source data are provided with this paper.

### Code availability
Code is available here: https://osf.io/eybc8.

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

## Acknowledgements

The authors thank Jing Lin and Jian Chen for programming the experimental stimulus. We also thank Eric Avila, Kaushik Lakshminarasimhan, Stefania Bruni, and Panos Alefantis for data collection, and Roozbeh Kiani for his surgical expertize during the Utah array implantations. The work was funded by 1U19 NS118246 (D.E.A.), 1R01 NS120407 (D.E.A.), 1R01DC014678 (D.E.A.) from NIH, and 1R01MH125571 (C.S.) from NIH, the National Science Foundation under NSF Award No. 1922658 (C.S.), and a Google faculty award (C.S). The work was also supported by K99NS128075 (J.P.N.). This work was also supported in part through the NYU IT High Performance Computing resources, services, and staff expertize.

## Author contributions

Cristina Savin and Dora Angelaki contributed equally to this work and are co-senior authors. J.P.N.: conceptualization, data curation, formal analysis, funding acquisition, investigation, methodology, visualization, writing - original draft, and writing - review & editing. E.B.: conceptualization, data curation, formal analysis, investigation, methodology, software, visualization, writing - original draft, and writing - review & editing. C.S.: conceptualization, funding acquisition, project administration, resources, software, supervision, and writing - review & editing. D.A.: conceptualization, funding acquisition, project administration, supervision, and writing - review & editing.

## Competing interests

The authors declare no competing interests.
