## [Peer Review File · Nature Communications]

Context-invariant beliefs are supported by dynamic reconfiguration of single unit functional connectivity in prefrontal cortex of male macaquesReviewer #1 (Remarks to the Author):

This paper uses modern "big data" methodology and a clever task to explore the substrates of goal-directed navigation in the primate brain. The paper is well written and concise, and clearly articulates the question under study, which is how an internal model of a dynamic scene is updated as an animal actively moves through a virtual world. This is the latest of a string of papers on the topic, and focuses on the dynamic correlations between neurons in two different contexts. It finds that these correlations change with the nature of the scene, and that these changes differ with the area being recorded. The only conclusions allowed by this design are of correlation, and in many places the authors are appropriately cautious in their conclusions, but at times their own beliefs influence the writing and the conclusions are over-stated. These data are novel and interesting, but I have some concerns about whether some features of the complex design might have been brushed aside, as well as about whether the novel analysis really reveals what they say it does.

Major concerns:

1) P-GAM model validation. Each neuron's data was fit with a model including a very large number of free parameters. The method was poorly described, and the previous methods paper on the subject (ref 30) was hard to follow for the non-specialist. Clearly the entire paper rests on this fitting, and neither this paper nor the previous attempt any sort of validation of the fitting using more traditional methods. While they say that their modeling produces confidence intervals for each parameter, the underlying assumptions are not well described. And in complex model fitting with non-orthogonal parameters, one worries about the weights shifting as one parameter stands in for another, which might even drop out of the model. So the authors need to better reassure the reader that the fitting produces results that can be trusted, especially the critical dynamic correlation terms between neurons. These are crucially important to the conclusions of the paper, but unfortunately hard to validate. There might be other ways, but one idea would be to isolate pairs of correlated neurons, identify trials with similar trajectories, and analyze the CCFs seen empirically in those trials. While the authors didn't mention it, one suspects they might have included identical "replay" trials (under fixation) in their design, and these would be perfect for validating the results.

2) The paper uses a clever 2D navigation task, yet in most of their analysis, they focus on only one dimension: the distance to the target, which they describe as "the critical task variable". But target azimuth is just as critical, and the animals make errors in both dimensions. In particular, the dispersion of responses seems to grow more in azimuth than in distance in the low-density condition (Figure 1D). This omission could be a problem in some places (e.g., the decoding, where correlated variance on the two dimensions could lead to misunderstanding), but could also be an opportunity to connect the results of this work to the historically much better studied questions of heading representation and perception.

3) The shuffle control is important, but is poorly described and possibly incomplete. The problem is that all the trials being shuffled within each neuron's data are not alike. While they are at the same (or similar) distance from the target, they could have gotten there by very different trajectories. So, their position on stimulus-variable tuning functions (e.g., time from start of movement) might be very different. Therefore, shuffling probably perturbs signal correlations as well as noise correlations. And that is all without considering the other dimension, angular velocity, which is only similar in the shuffled trials by virtue of statistical correlations in the behavior.

4) Eye movements. Previous work using this task, by some of the same authors, has shown the importance of eye movements in the performance of this task. Indeed, in that paper (ref 25) the eye movements are clearly correlated with the monkey's sense of where it is in the virtual world, and might even carry that information explicitly. Also, ongoing and upcoming eye movements have been documented to affect neurons in all three areas under study in this paper. Eye position is included in the model, but eye velocity is not. And neither is mentioned in the text. Given that pursuit signals in MSTd have nonlinear dynamics (Krauzlis and Lisberger), a linear position signal will probably not capture much of it. Common eye movement signals could then easily masquerade as noise correlation.

5) The main interpretation of the paper rests on stability of dynamics being related to changes in

coupling. They nicely document the changes (or lack thereof) in coupling, but do not show anything about the stability of dynamics or tuning across conditions. I'd like to see something akin to Fig. 2D for neuronal tuning functions, split out by the stimulus variable. Additional strength could be brought to their favored interpretation if neurons with larger numbers of coupled partners were more stable in their tuning.

6) With regard to interpretation, the authors attach a lot of importance to the coupling, but it is relatively rare. How much influence can it have? Since we have no quantitative estimates of coupling strength (in terms of spike rates), the reader is at a loss. I think it would vastly help their conclusions if they could give the coupling strength in more interpretable units. I don't know if it is possible in their modeling, but one manipulation they should consider is to implement the model artificially, then remove all the coupling terms and observe the stability of the network in each area.

7) The coupling filters are very short, and all the examples show them to be highly asymmetrical. This does not resemble the coupling between neurons measured with more traditional methods, which is often rather extended in time.

Smaller concerns:

8) In their description of the task, they say that the texture elements can't be used as landmarks. This is technically true, but is a little disingenuous, since the monkeys have several cues available to be used for landmark-based navigation. These include texture element size and density, and target location on the screen.

9) The methods need to be much better described. There is no mention of number of trials or spikes for each of the neurons, nor of the number of neurons and how repeated neurons across sessions were treated. More descriptions of the assumptions and limitations of the fitting procedure also need to be given for the non-specialist audience.

10) p. 6, para 3. Description of gain analysis is confusing, and might be wrong. The legend to Figure S3C talks about using a scale factor to estimate a gain change, which could be correct, but the main text confusingly talks about using "the slope of this correlation" to make the same estimate. Surely there's a more straightforward way to estimate gains, or to describe their approach.

11) p. 6, last paragraph. The phrase "...as if attempting to offset the unavailability..." makes little sense to me, and the costs and benefits of coupling depend on details of signal and noise that are unavailable. In any case, it is highly speculative and doesn't belong in the results section.

Reviewer #2 (Remarks to the Author):

In this manuscript, Noel et al study the neural mechanism underlying context invariant computations, i.e. the ability of animals to generate similar behaviors in settings characterized by different sensory inputs. The authors record neural population responses from visual, parietal, and prefrontal areas in monkeys engaged in a task requiring them to navigate with a joystick towards a remembered target location. The monkeys perform this task in two contexts, one in which they can infer their position in the environment based on strong visual cues, and one where the visual cues are impoverished, thus forcing the monkeys to rely more on learned internal models capturing the relation of joystick inputs and the resulting movement within the environment.

The authors analyze population responses with two complementary approaches that are typically not considered at the same time: (1) a new framework they call generalized additive model, similar to GLM, in particular to infer the nature of coupling between neurons that is responsible for "noise correlations"; (2) low-dimensional population trajectories and decoding. These approaches reveal several potentially interesting features of population responses in the recorded areas. One

main finding is that context-invariant dynamics (at the level of trajectories and decoding) seems to correlate with a context-dependent "remapping" of the coupling between neurons. This relation is evident in a comparison across areas, as well as across sessions in PFC.

I find the general question and methods timely and important. The results are intriguing and potentially of broad relevance. However, the authors should perform additional analyses and controls to help in the interpretability of their findings, and/or to establish a more direct relation to past models of context-dependent computations

Main comments:

(1) In the description of the behavior, I did not find any explanation of the eye movements of the monkeys during the task. Are monkeys fixating? Or are they free to move the eyes? Based on their REF 35, I assume the eyes are free to move. If so, it seems imperative to compare eye movements across contexts, as such movements could have a strong impact on neural responses in the recorded areas.

(2) Several features of the GAM are not explicitly justified in this manuscript (although they may have been addressed in previous papers by the authors employing similar methods), and it remains unclear how somewhat different formulations of this model would change the results. Addressing the points below seems important to interpret the reported findings.

(2a) The model includes a number of external and latent variables, but some prominent variables could arguably have been added. For example, variables reflecting sensory or latent variables in retinotopic coordinates; a variable encoding context (strong or weak optic flow); a variable encoding the passage of time in a trial (which appears to be encoded e.g. in PFC, based on the data in this paper and based on past research); or variables encoding eye movements beyond the horizontal and vertical gaze location (for example eye velocity, which would encode saccades and could lead to strongly correlated firing across neurons). How would the results change if these variables were included?

(2b) The GAM includes both filters implementing tuning for particular variables (like distance to target) as well as temporal filters (spike history and coupling). But it seems that the tuning filters are not directly linked to a "temporal" filter. Does this mean that these variables have instantaneous effects on neural responses? In many areas, the onset of a stimulus for example leads to a somewhat transient activation. How would such a temporal effect be captured in this model?

(2c) In this paper, the authors do not show any evidence that their method can retrieve parameters of a ground truth model ("model recovery"). The authors also do not explore cases of "model mismatch", where the structure of the fitted model does not match that of the ground truth. For example, due to failure to include some input variables that are present in the ground truth, but are not included in the fit; or inclusion of variables that are correlated with the ground truth variables, but not identical to them. Generally, it would be comforting to see that the results are somewhat robust to such mismatch.

(3) The authors report several potentially interesting relations between insights from their GAM model (e.g. how neuron couplings change across contexts) and population trajectories/decoding (e.g. whether representations are context-invariant or not). These relations are interpreted in terms of past models of context-dependent computations, but some of the related conclusion would be stronger with some additional analyses.

(3a) The authors compare task representations and their context-invariance at the level of population trajectories. But could one also employ the GAM stimulus filters? For example, does invariance of average trajectories imply that the stimulus filters should also be invariant? Did the authors evaluate the similarity of stimulus filters across contexts in the same way as they did for coupling filters?

(3b) Artificially eliminating correlations results in distance to target not being decodable anymore

across contexts in PFC. I found this result surprising, as the trajectories in Fig. 3A seem to show a component of the dynamics that is (1) invariant across context; (2) encodes distance to target. Distance appears to be encoded non-linearly, as a combination of distance from the "origin" and angle within the 2d-plane (the rotation from the bottom right to the top left). Would a different, non-linear decoding approach (e.g. an artificial neural network) result in good decoding even after removing correlations?

(3c) The interesting relation in Fig. 3E (dIPFC) is shown only for distance to target. I understand the importance of that variable, but according to the GAM many other variables are also represented, and one could equally ask if their representation becomes more task invariant as a function of the remapping of coupling filters. Does decoding of those variables show a similar relation?

(3d) Could the authors explore the relation of stimulus filter, coupling filters, and dynamics in previously proposed recurrent neural network models of context-dependent computations? Analyzing such RNN would probably require some changes in the definition of the GAN (as the RNN units are not spiking) but potentially an account of variability in terms of stimulus filters and coupling filters would nonetheless appear meaningful. If such an analysis was possible, it would help in establishing a connection to the past work that the authors use to frame their results.

Additional comments:

(4) "The slope of this correlation quantifies a change in the gain of neural responses driven by task-relevant stimuli".

This is true only if the tuning stays the same? If the tuning changes, this measure can be zero, even though the gain has changed a lot.

(5) "While the location of mean endpoints did not change across densities".

Not clear how this relates to the observation in Fig. 1C that the radial bias seems to change across contexts. Would a change in bias not result in a change of the means?

Reviewer #3 (Remarks to the Author):

Noel, Balzani, and colleagues.

This is an interesting paper that contains a large amount of new data drawn from three cortical regions – in parietal, prefrontal, and extrastriate visual cortex – in the macaque as they adjust to changes in sensory context. Changes in interneuronal coupling patterns in the dIPFC and 7a occur as the sensory context changes and in dIPFC these changes are predictive of a stable population code and stable behaviour. I think that the results will be of broad interest. However, the brevity of the format means that sometimes some points are difficult to understand and some additional clarification would be helpful.

1. Could you clarify the angular bias and radial bias indices discussed on page 5 and illustrated in figure 1c? What is the bias towards here? The mean angular and radial position across trials? Or something else (such as the last trial). Do these perhaps relate to the section on page 16 that says "A linear model with multiplicative gain accounted well for the observed data (average $R^2 = 0.72$). Thus, we used the slopes of the corresponding linear regressions as a measure of bias" ? If so, maybe the reader could be guided to this section of the Methods? What slopes are being taken from the linear regression? What are they related to? It is not clear why they constitute a measure of bias; could this be explained?

2. It is difficult to reconcile figure 2d and the text describing it at the top of page 7. The top of page 7 describes MSTd as showing the least remapping of inter-unit correlations but the figure 2d shows the green line as the line closest to the "maximum" line. Some more explanation of what is

being computed and what is being illustrated would be helpful.

It would be especially helpful to have a sense of what the "cumulative density" is of and how it is calculated. I think that the word "density" here is not related in a simple way to the low density optic flow and high density optic flow conditions. Is that correct? Why are the data being plotted in this way? Given the importance of this figure to the authors' claims, it would be useful to have a more intuitive explanation of what is being shown here.

3. Is it possible to show readers where the recording sites were? Or, if that is not possible, could more verbal description be given? While the MSTd might be a relatively well defined region, the borders of some of the others are treated differently by different researchers. For example, dIPFC is used to refer to quite a wide region of tissue from the periarculate area to considerably further forward, from within principal sulcus to dorsal to dorsal convexity.

P3 "sufficient to identity..." should probably be "sufficient to identify..."

P11 typo in "input to the system are state-dependent and continuously changing"

Reviewer #1 (Remarks to the Author):

This paper uses modern “big data” methodology and a clever task to explore the substrates of goal-directed navigation in the primate brain. The paper is well written and concise, and clearly articulates the question under study, which is how an internal model of a dynamic scene is updated as an animal actively moves through a virtual world. This is the latest of a string of papers on the topic, and focuses on the dynamic correlations between neurons in two different contexts. It finds that these correlations change with the nature of the scene, and that these changes differ with the area being recorded. The only conclusions allowed by this design are of correlation, and in many places the authors are appropriately cautious in their conclusions, but at times their own beliefs influence the writing, and the conclusions are over-stated. These data are novel and interesting, but I have some concerns about whether some features of the complex design might have been brushed aside, as well as about whether the novel analysis really reveals what they say it does.

Major concerns:

1) P-GAM model validation. Each neuron’s data was fit with a model including a very large number of free parameters. The method was poorly described, and the previous methods paper on the subject (ref 30) was hard to follow for the non-specialist. Clearly the entire paper rests on this fitting, and neither this paper nor the previous attempt any sort of validation of the fitting using more traditional methods. While they say that their modeling produces confidence intervals for each parameter, the underlying assumptions are not well described. And in complex model fitting with non-orthogonal parameters, one worries about the weights shifting as one parameter stands in for another, which might even drop out of the model. So the authors need to better reassure the reader that the fitting produces results that can be trusted, especially the critical dynamic correlation terms between neurons. These are crucially important to the conclusions of the paper, but unfortunately hard to validate. There might be other ways, but one idea would be to isolate pairs of correlated neurons, identify trials with similar trajectories, and analyze the CCFs seen empirically in those trials. While the authors didn’t mention it, one suspects they might have included identical “replay” trials (under fixation) in their design, and these would be perfect for validating the results.

We thank the reviewer for this question, which is indeed critical for the rest of the conclusions. In response to this comment, we have:

- (1) Fit the model allowing or not for coupling functions, and correlated the shape of tuning functions in the coupled vs. uncoupled model. If the coupling functions are accounting for something else rather than stimulus-independent neural co-fluctuations, then the tuning functions (i.e., “signal correlations”) should differ in coupled and uncoupled models.
- (2) Fit the original model multiple times (different initializations) while artificially zeroing out components estimated to impact a neuron’s firing rate. We parametrically vary the number of components we zero out, starting from the component deemed to most impact neural responses (according to mutual information) and subsequently zeroing out the 2nd, 3rd, etc. components. We zero out up to 10 variables the model deems to impact a neuron’s firing rate. Then, we compare the estimated tuning and coupling functions of non-zeroed variables, via correlations (akin to many of the analyses conducted in the main text). In other words, we empirically test whether as the reviewer suggests “weights shift as one parameter stands in for another” by excluding significant variables and examining the shape of the remaining functions (signal and noise).
- (3) Isolate pairs of neurons the model estimates as functionally coupled (or not, as a control) and derive an empirical spike-triggered average – as the reviewer suggests.
- (4) Fit a new PGAM model with different inputs from that originally reported. Namely, we include eye velocity in horizontal and vertical planes (see comments below). Further, we also include the instantaneous distance/angle to target and distance/angle to the animal final stopping position (reflecting its belief of where the target is) *in screen (or “egocentric”) coordinates as opposed to virtual 3-D coordinates*. We then examine whether we replicate the central finding from the original manuscript (noise correlations change in 7a and dIPFC more than in MSTd). This analysis allows testing for the robustness of our main result, testing if and how sensitive it is to the particular encoding model used.

- (5) Fit the new PGAM encoding model (described above in point 4) and then perform correlation on the estimated tuning functions, in order to ascertain how dependent these tuning estimates are on the specific model inputs.

In the following paragraphs (below) we present the results of these new analysis. However, prior to presenting these results, we must also highlight that contrary to what the reviewer stated, we have previously extensively validated the modeling approach (Balzani et al., 2020 and Noel et al., 2021), including by using “more traditional methods”. Namely, previously we have:

- Generated spike trains according to real input statistics (e.g., linear and angular velocity) while including or excluding (i.e., zeroing out) a single variable, and then determine whether the model appropriately selected or excluded this component. We showed 0.0007% false positive rate, and 0% false negatives (Figure 2, supplement 2; Noel et al., 2021, *eLife*). Here, we extend on this analysis by zeroing out more than a single variable (i.e., up to 10), and by examining the correlation between the resultant shape of tuning functions (as opposed to the binary; significant or non-significant).
- Visualized raw data for neurons the model estimated to show spike-LFP phase coupling (Figure 2, supplement 8; Noel et al., 2021, *eLife*) and estimated this spike-LFP phase coupling via pairwise phase consistency (Figure 2, supplement 9; Noel et al., 2021, *eLife*; Vinck et al., 2010). The results from this traditional approach confirmed the results from the PGAM.
- Estimated the fraction of neurons tuned to different task variables in odd vs. even trials, demonstrating very stable estimates (Figure 2, supplement 10; Noel et al., 2021, *eLife*).
- Included nuisance variables strongly correlated ($r = 0.7$; average correlation in real input data is ~ 0.15) with the true covariate but not driving neural responses. The model correctly rejected *all* nuisance variables even with 5 minutes of data, and had no misses as long as it had access to 20+ minutes of data (Balzani et al., 2020, *NeurIPS*). Our recording sessions here are on the order of 180 minutes/session.
- Showed that the model recovers well established properties of functional connectivity, namely that neurons show “like-to-like” connectivity (Chettih & Harvey, 2019, *Nature*; **Figure S1B** in the current manuscript), as well as stronger coupling to nearby neurons (Rosenbaum et al., 2017, *Nature Neuroscience*; **Figure S1A** in the current manuscript; also see Oldenburg et al., 2024, *Nature Neuroscience*, for a recent demonstration of these same effects using ensemble-specific two-photon optogenetics in the mouse visual cortex).

Altogether, we conclude that the modeling approach (under appropriate data regimes) is very robust. In fact, this model is now employed by other groups (e.g., Drs. Nachum Ulanovsky and György Buzsáki) and is being implemented by the Flatiron Institute as a standard (and open access) framework for statistical modeling in systems neuroscience.

(1) Signal correlations are robust across coupled and uncoupled models: coupling filters do not stand in for other parameters.

We directly test whether coupling filters (i.e., noise correlations) are “standing in” for other variables by performing correlations between tuning functions (same neuron, trials, session, etc.) estimated with and without coupling functions. The latter model has approximately 17% of the number of parameters than that reported in the main text (uncoupled model ~ 270 parameters; coupled model ~ 1581 parameters). As shown in **Figure S6D**, the tuning function are unaltered by the inclusion of coupling filters, being correlated at $\sim r = 0.93$ on average (note the range on the y-axis, from 0.85 to 1.0). There is no difference in signal correlation across brain areas in coupled vs. uncoupled models (one-way ANOVA, $p = 0.88$).

Figure S6. Coupling filters account for inter-neuron correlations. **A.** Example tuning, spike-history, and coupling filters estimated by the P-GAM. From top to bottom: Raw (gray) and P-GAM reconstructed (black) filters for task-variables (demonstrating the model's ability to account for spiking activity), spike history filters (showing the characteristic refractory period of single units), and coupling. **B.** Pseudo- R^2 for uncoupled (x-axis) and coupled (y-axis) models, showing that P-GAM encoding models allowing for neuron-to-neuron coupling accounted better for observed spike trains. Each dot is a neuron and the identity line in red (note different range in x and y). **C.** Five snippets (each 30 seconds long) of "observed" neural activity (convolved with a Gaussian filter with a standard deviation of 20ms, black), as well as predicted firing rates with a P-GAM allowing for coupling (coupled model, red), or not (uncoupled mode, blue). **D.** Correlation between tuning functions to different task variables (x-axis) estimated with the coupled and uncoupled model (y-axis = r-values, average $r = 0.93$, all variables above $r = 0.85$). The different brain areas are colored (MSTd = green; 7a = blue, dlPFC = red), and there was no difference in the stability of these tuning functions across brain area (one-way ANOVA, $p = 0.68$). Note, however, there was a difference in their gain-modulation, being most readily modulated in MSTd than 7a or dlPFC (see **Fig. 2B**). **E.** Example (12) across-units spike-triggered averages (i.e., average firing rate of "receiver" neuron condition on a spike from a "sender" unit), as empirically estimated (black) and estimated via coupled (red) or uncoupled (blue) P-GAM encoding models. Ten of these neurons (all but bottom row, 3rd and 4th from the left; transparent colors) were estimated as coupled by the P-GAM, while the last two (shown as a negative control) were determined not to be coupled. Of note, the coupled model (red) is able to recapitulate neuron-to-neuron dynamics, while the uncoupled model is not (mean R^2 , coupled model, 0.65 ± 0.02 ; uncoupled model, $R^2 = 0.26 \pm 0.06$; $p = 0.0003$). Further, we determine 95% CIs for each sender-receiver pair by shuffling spike times of the "sender" unit 1000 times and re-computing spike-triggered averages. On average, the P-GAM estimated 11.31% of unit pairs to be coupled, while this number was much higher (39.8%) according to permutation testing on raw data. This discrepancy highlights the critical need for an encoding model, accounting and explaining away signal-correlations. In other words, while the empirical spike-triggered averages may be driven by both neurons responding to a "signal", the model estimated pairs are conditioned on this other input, and thus results in a more conservative (i.e., statistically robust) estimate. When both the coupled model and empirically-estimated spike triggered averages agreed on indexing a coupled pair, their R^2 was 0.75. In conclusion, the real benefit to the coupled P-GAM is not in better accounting for spike times of a single neuron (**B** and **C**) but in being able to account for neuron-to-neuron dynamics above and beyond signal correlations (**E**, also see **Figure 4C** in Lakshminarasimhan et al., 2023).

We must also highlight that the coupling filters do add significantly to our ability to explain variance. **Figure S6B** shows the pseudo- R^2 for uncoupled (x-axis) and coupled (y-axis) models (each dot is a neuron fit, red line is the identity line). The coupled model accounts for 1.55 times the variance accounted for by the uncoupled model.

(2) Signal and noise correlations are robust to zeroing out other (significant) components.

We test the hypothesis that variables are standing in for one another by simulation. For each neuron, we re-fit the coupled encoding model, while artificially zeroing out other components that the full model deemed significant in accounting for variance. More specially, we examine the correlation between tuning functions to target distance (left-most panel), movement offset (second panel), linear velocity (third panel), as well as noise correlations (right-most panel), when zeroing out other significant contributors. The neurons examined are all significant for the variables mentioned (target distance, movement onset, linear velocity, coupling functions), and we zero out other variables (e.g., linear acceleration) in order of mutual information (akin to effect size in contributing to estimating firing rates). As shown in **Figure S2**, the tuning and coupling functions remain very stable (~ range = 0.9 to 1.0). Importantly, there is also no evidence that tuning function correlation becomes unstable after a larger number of variables have been removed (one-way ANOVA, all $p > 0.79$). As such, it appears that the encoding model is appropriately isolating tuning and coupling functions, without parameters standing in for one another.

Figure S2. P-GAM validation against ground-truth data. To test for the ability of our encoding model (P-GAM) to recover ground-truth data within the statistical regime of the current dataset, we artificially zeroed out variables that a first full fit deemed to significantly account for variance, and then fit again. More specifically, we sequentially zero out 1 to 10 significant task variables, in order of mutual information (from most to least). On no occasion (0.00%) did a neuron originally deemed to be tuned to either distance to target, movement offset, linear velocity, or the spiking of other neurons (i.e., coupling filters), was it deemed not to be tuned to this variable in re-fits (>20k re-fits). Similarly, on no occasion (0.00%) was a neuron not originally estimated tuned to one of the above-mentioned variables, considered tuned on follow-up fits (see REF³⁰ for a similar analysis demonstrating no mis-categorizations when recordings are over 20 minutes long. Here they are ~ 180 minutes long). Now, we can be more stringent, and not only examine if neurons change categorization from tuned to not-tuned (or vis-versa) while zeroing out other variables, but we can also examine the shape of the resultant tuning functions. Each time, we examine the correlation between the original and re-fitted tuning functions to target distance (left-most panel), movement offset (second panel), linear velocity (third panel), as well as noise correlations (right-most panel), while zeroing out other variables (e.g., linear acceleration). The tuning and coupling functions remain very stable (~ range = 0.9 to 1.0). Importantly, there is also no evidence for the fact that tuning function correlation becomes unstable after a larger number of variables have been removed (one-way ANOVA, all $p > 0.79$). As such, it appears that the encoding model is appropriately isolating tuning and coupling functions, without parameters standing in for one another.

(3) Coupled encoding models can account for empirical spike-triggered averages. Uncoupled models cannot.

We compute empirically-observed spike-triggered averages. That is, each time a ‘sender’ unit spikes, we epoch the 1.2 seconds after this spike in a potential ‘receiver’ unit. We then compute spike-triggered averages (akin to a cross-correlation). We also perform this computation another 1000 times, while shuffling the timing of the ‘sender’ unit spikes, in order to create a null distribution. In **Figure S6E** (see above) we show the empirically derived spike-triggered average, as well as the firing rate estimates from a coupled (red) and uncoupled (blue) model, for 12 example neurons. Ten of these neurons (all but bottom row, 3rd and 4th from the left; transparent colors) were estimated as coupled by the P-GAM, while the last two (shown as a negative control) were determined not to be coupled. The coupled model (red) is able to recapitulate neuron-to-neuron dynamics, while the uncoupled model is not (mean R^2 , coupled model, 0.65 ± 0.02 ; uncoupled model, $R^2 = 0.26 \pm 0.06$; $p = 0.0003$). This is the real benefit of including coupling filters in the P-GAM; to explain neuron-to-neuron correlations (also see Figure 4C in Lakshminarasimhan et al., 2023).

On average, the P-GAM estimated 11.31% of unit pairs to be coupled, while this number was much higher (39.8%) according to permutation testing on raw data. This discrepancy highlights the critical need for an encoding model, accounting and explaining away signal-correlations. In other words, while the empirical spike-triggered averages may be driven by both neurons responding to a “signal”, the model estimated pairs are conditioned on this other input, and thus results in a more conservative (i.e., statistically robust) estimate.

When both the coupled model and empirically-estimated spike triggered averages agreed on indexing a coupled pair, their R^2 was 0.75.

(4) An alternative encoding model replicates the remapping of coupling filters in 7a and dIPFC

To further test the robustness of our main result (remapping of coupling filters in 7a and dIPFC) we questioned if this effect was sensitive to the precise inputs to the encoding model. Thus, we fit all data to an alternative encoding model. Instead of expressing the distance to target in a virtual three-dimensional space (i.e., with depth), we express distance to target as a vertical and horizontal angle between (1) the projection of the target onto the screen, (2) the eyes socket, and (3) the current gaze position on the screen. Further, instead of including the radial and angular distance the animal has path integrated, we include the (egocentric) distance remaining to its stopping position. Note, distance to target and from origin are correlated on a single trial, but are not across trials (given targets are presented at a random distance). We include distance to stop, as the animals are trained to stop where they believe the target to be positioned, and thus their stopping location is a proxy for their (subjective) belief of where the target is. Lastly, we also include in this model the vertical and horizontal eye velocity (not included in the original model; also see Question #4). The elements modified in this alternative encoding model are shown in orange in **Figure S9A**. The fraction of neurons tuned to the different experimental variables are shown in **Figure S9B**. Overall these results replicate those from **Figure S7A** and Noel et al., 2022, *eLife*, with the most noteworthy change being the fact that more neurons are tuned to eye velocity than position. Most importantly, we replicate the fact that area 7a (blue) and dIPFC (red) more readily remap their noise correlations as a function of density than MSTd does (green; **Figure S9D**). This results highlights that the differential remapping of coupling filters is robust across encoding models.

Figure S9. An alternative encoding model reproduces the dynamic remapping of noise correlations in 7a and dIPFC. **A.** Alternative encoding scheme (contrast with Fig. 2A, differences highlighted in orange here). This alternative P-GAM expressed distances to target in egocentric coordinates. Namely, instead of conceiving the targets in a three-dimensional space with depth, the targets are now projected onto the two-dimensional coordinates of a screen. Instead of a radial and angular distance to target, now the targets are expressed by a vertical and horizontal angle vis-à-vis the ongoing location of gaze (also on the screen). Further, we test the hypothesis that animals may not path integrate distances from origin, but from their eventual stopping locations, and thus we include egocentric distances (vertical and horizontal) to the eventual stopping location. Lastly, we include not only eye positions, but also eye velocity. **B.** Fraction of neurons tuned to the different variables in the alternative encoding model. These are congruent with prior results (REF³⁵ and Fig. S7A), showing a patterned mixed selectivity, with greater coding for sensorimotor variables in 7a and of latent variables of dIPFC. Further, the results show greatest coding of eye velocity in MSTd. **C.** Signal correlations (r-value between tuning functions estimated in the different encoding models) are a function of brain area (MSTd in green, 7a in blue, and dIPFC in red) and task variable (restricted to the subset of variables present in both the main-text encoding model and the alternative one presented here). **D.** Coupling filter correlation (r-value in shape of coupling function in low- and high-density optic flow condition) as a function of brain area (MSTd in green, 7a in blue, and dIPFC in red) and encoding schema (main text on the left, and alternative P-GAM on the right). The findings replicate the results demonstrating greater remapping in 7a and dIPFC as opposed to MSTd with the change of optic-flow density. Error bars are ± 1 S.E.M.

(5) Robust estimation of signal correlations across encoding models

We examine the robustness of the statistical approach we employ by examining the correlation between tuning functions estimated across encoding models (encoding 1 = main text; encoding 2 = alternative encoding model introduced above). Given that a particular neuron was estimated as tuned to a particular variable in both encoding

models (99.1% of cases), we performed a Pearson correlation between estimated tuning functions. Note, we do this analysis for 10 variables, those overlapping across encoding models and not being circular in nature (i.e., phase of LFP). The results are shown in **Figure S9C**, demonstrating once again very high r-values across encoding models (mean of all variables $r \sim 0.92$).

Conclusions

Altogether, across a number of new analysis (presented above) and prior validation of the model (Balzani et al., 2020, *NeurIPS*; Noel et al., 2022, *eLife*), we conclude that the model is robust, with tuning functions being very similar across encoding models used and independent of whether coupling filters are included or not. There is very little (if any) “bleeding” of parameters from one variable to another, as estimated by simulations in which we artificially zero out variables impacting neural firing. This is likely not only a feature of the modeling approach, but also the fact that we use a naturalistic experiment wherein no two trials are alike, and most variables are continuous/“analog” signals (as opposed to “binary” in 2 alternative forced choice experiments); thus the statistical regime of our data allows for robust estimation of tuning functions.

We have a very limited set of “replay” trials, and only in the high-density condition. Further, while the baseline firing rates do not change as a function of whether the animals are engaged or not in a task, their modulation by task variables does (see Figure 2 – supplement 11 in Noel et al., 2022, *eLife*).

In addition to including **Figures S2, S6, and S9** (and their associated figure captions), we have also modified the text in the following manner:

“This technique recapitulates known properties of noise correlations, such as its dependency on the distance between neurons³³ and tuning similarity^{20, 34} (Fig. S1, also see³⁰ and Fig. S2 for further model validation, and REF³⁵ for a detailed description of spiking responses and spike-LFP phase coupling during optic-flow guided navigation, i.e., the “high-density” condition).”

And:

“The P-GAM accounted well for spiking activity (average pseudo- $r^2 = 0.091$, ~ 1.5 to 2 times better than traditional Generalized Linear Models³⁸, see Fig. S6A for example tuning functions, both as raw binned responses and as estimated by the P-GAM), and did so better when including (vs. excluding) coupling filters (~ 1.5 times better in pseudo- r^2 and ~ 1.14 times in R^2 , Fig. S6B and C). Importantly, the addition of coupling filters did not change the estimate of tuning functions (correlations between tuning functions estimated with and without coupling functions, $r \sim 0.93$, Fig. S6D), but critically, did allow to additionally account for signal-independent inter-neuron correlations (model prediction vs. empirical spike-triggered average, coupled model, $R^2 = 0.65 \pm 0.02$; uncoupled model, $R^2 = 0.26 \pm 0.06$; $p = 0.0003$; see Fig. S6E for examples and caption for detail).”

And:

“This analysis showed that the temporal structure of stimuli-independent neural co-fluctuations partially remapped with context in all areas (greater than chance, smaller than max; Fig 2D shows full distributions as cumulative probabilities, see Fig. S9 for a different visualization), but did so less in MSTd (average r , 0.85 ± 0.017) than in 7a (0.73 ± 0.002) and dIPFC (0.73 ± 0.014 , Kruskal-Wallis contrasting areas, $p = 0.0021$; post-hoc Holm-Sidak corrected contrasts, MSTd vs. 7a, $p = 0.003$; MSTd vs. dIPFC, $p = 0.02$; 7a vs. dIPFC, $p = 0.76$, Fig. 2D). To test for the generalizability of this finding, we fit an alternative encoding model wherein distances to target were expressed in egocentric coordinates, and we included distances to the eventual monkey stopping location as opposed to distance from origin. Further, we included regressors accounting for not only eye position, but also eye velocity (Fig. S9A-C). We confirm that also with an alternative encoding schema, areas 7a and dIPFC remapped noise correlations more readily than MSTd (Fig. S9D, Kruskal-Wallis, $p = 0.0023$).”

2) The paper uses a clever 2D navigation task, yet in most of their analysis, they focus on only one dimension: the distance to the target, which they describe as “the critical task variable”. But target azimuth is just as critical, and the animals make errors in both dimensions. In particular, the dispersion of responses seems to grow more

in azimuth than in distance in the low-density condition (Figure 1D). This omission could be a problem in some places (e.g., the decoding, where correlated variance on the two dimensions could lead to misunderstanding), but could also be an opportunity to connect the results of this work to the historically much better studied questions of heading representation and perception.

We thank the reviewer for this question and the opportunity to clarify a misunderstanding. The radial distance to target is indeed the only task-relevant latent variable; animals are rewarded for stopping within 65cm of the target. This measure of radial distance to target includes both depth and azimuth, given that we are computing *radial* distances; radial distance to target = $\sqrt{\text{distance along azimuth}^2 + \text{distance along depth}^2}$.

The angular distance to target is an intuitive variable when the target is first presented (e.g., “the target is at 45 degrees from straight ahead”), but the evolving angle between target and animal (angular distance to target) is (1) inconsequential to the animal, given that it will be rewarded for any angle it stops from the target as long as the radial distance is small enough – the reward zone is a circle – and (2) it is a very volatile/noisy variable, particularly as the macaque gets close to the target. For instance, undershooting the target by 1 cm, versus overshooting it by 1cm (2cm difference), will result in a change of the angle to target of 180 degrees.

Lastly, we wish to point out that angular variables are explicitly included in most analysis (and implicitly included in the rest, such as decoding distance to target which includes both depth and azimuth), as well as explored in detail in Noel et al., 2022, *eLife* (which is based on the high-density condition). The central interest here is in determining how the task-variable is encoded/decoded and how this differs across density conditions requiring different computational approaches.

We have modified the text in the following manner in order to clarify this issue:

“Next, to quantitatively test the hypothesis that the manifold coding for distance to target – as opposed to population dynamics as a whole (as in PC space - Fig. 3A, Fig. S10) – would remain stable in 7a and dIPFC, we used a linear population decoder (Lasso regression, 5-fold cross-validation) to estimate the animal’s evolving distance to target. Note, this distance to target is the key latent task variable leading to reward ($\rho < 65\text{cm}$, see Methods) and accounts for evolving position relative to target along both azimuth and depth (distance to target = $\sqrt{\text{distance along azimuth}^2 + \text{distance along depth}^2}$). Instead, animals can be rewarded for any angular distance to target (the target and reward zone being circles).”

3) The shuffle control is important, but is poorly described and possibly incomplete. The problem is that all the trials being shuffled within each neuron’s data are not alike. While they are at the same (or similar) distance from the target, they could have gotten there by very different trajectories. So, their position on stimulus-variable tuning functions (e.g., time from start of movement) might be very different. Therefore, shuffling probably perturbs signal correlations as well as noise correlations. And that is all without considering the other dimension, angular velocity, which is only similar in the shuffled trials by virtue of statistical correlations in the behavior.

We thank the reviewer for this important question. We address it in six ways. First, we empirically checked how perturbed task variables were by our shuffling procedure. As it can be appreciated in the figure below (**Figure S11**, while distance to target is well matched between shuffled and non-shuffled data (by construction), the remaining variables are also fairly well behaved, $r > 0.2$ (**Figure S11A** shows an example trials, and **Figure S11C**, x-axis, shows the correlations between shuffled and unshuffled data for all task variables across all sessions).

Second, we construct a more stringent shuffling procedure where in addition to matching distance to target, we now also match linear velocity. As it can be observed in **Figure S11B** (examples) and **Figure S11C** (y-axis), now (by construction), linear velocity also matches very well across shuffled and unshuffled data. The other variables also become slightly better correlated (all $r > 0.3$). The effect we report in the main text (i.e., inability to cross-decode in dIPFC) is robust to this new shuffling procedure (**Figure S11D**).

Figure S11. Artificially removing noise correlations. To test for the impact of noise correlations in allowing for distance to target cross-decoding, we match time epochs with the same distance to target (and thus output from the tuning function) and shuffle neural activity like-for-like. Meaning, we conserve signal correlations while eliminating the naturally occurring noise correlations. **A.** Examples showing “unshuffled” (black) and shuffled (red) task variables, when matching only distance to target. The correspondence between unshuffled and shuffled distance to target is very high by construction. The rest of variables (e.g., linear velocity, eye positions) are moderately correlated given the correlation between variables in this naturalistic task (grand average $r \sim 0.15$). **B.** For a more stringent test (vs. main text) here we additionally match linear velocity. Now distance to target and linear velocity are very well matched, by construction. The rest of variables also become slightly better correlated. **C.** Scatter plot (error bars are ± 1 S.E.M.) showing the average correlation between task variables in shuffled and unshuffled conditions, when matching solely distance to target (x-axis), or both distance to target and linear velocity (y-axis). In the latter case, all task variables are correlated in shuffled and unshuffled conditions above $r = 0.3$. **D.** Decoding of distance to target (CV R^2 , y-axis) in high-density (HD/HD), low-density (LD/LD), or across contexts (HD/LD) when eliminating noise correlations and matching for both distance to target and linear velocity. As in the main text (distance to target matching), across context decoding was abolished by the lack of noise correlations in dIPFC (red) but not in 7a (blue). MSTd (green) cannot cross-decode even with noise correlations (see main text).

We do not extend this procedure to a 3rd, 4th, 5th, etc. variable, as each time the number of data-points per 'cell' (e.g., match distance x velocity matrix) is reduced by approximately an order of magnitude, and thus matching further variables would not allow us to perform decoding within a robust statistical regime.

Third, we examine the specificity of the effect reported: the fact that distance to target is decodable across contexts in dIPFC (as in 7a), but that this capacity is impacted by the loss of noise correlations (in contrast to 7a). Namely, given that neural activity in a given area (MSTd, 7a, or dIPFC) allows for linear cross-context decoding of a particular task variable (e.g., linear velocity, angular distance, horizontal eye-position), we examine if and how eliminating noise correlations impacts the decodability of this variable. As in can be appreciated in **Figure S12A**, the ability to decode distance to target in dIPFC was the only task variable/area to become not different from chance (i.e., 95% CIs overlap with $CV R^2 = 0$) when excluding noise correlations.

A - linear decoder

B - non-linear decoder (ANN)

Figure S12. Decoding task variables within and across contexts, including and excluding noise correlations. A. Decoding cross-validated R^2 for each continuous task variable (panels) and brain area (MSTd = green, 7a = blue, dIPFC = red), both within contexts (HD/HD, LD/LD) and across contexts (HD/LD). Squared filled markers are used to indicate decoding including noise correlations (i.e., “unshuffled”) while empty circles are used to indicate decoding excluding noise correlations. The latter are only plotted when the decoding for a specific area/task variable is significant when including task variables. Error bars are $\pm 95\%$ CI, and thus non-overlapping error bars with CV = 0 (dashed line) is significant. The only brain area/task variable that is decodable across context with noise correlations but not without is distance to target in dIPFC. **B.** Same as **A**, while employing a non-linear artificial neural network for decoding. Results are replicated.

Fourth, we ascertain whether the effects reported would replicate to another decoding approach. Thus, we build a non-linear 2-layer ANN (ReLU rectifications, input dimensions 10x20). As seen in **Figure S12B**, we replicate the fact that only distance-to-target, and only in dIPFC, is cross-context decoding statistically eliminated by elimination of noise correlations.

Fifth, we want to point out that the reviewers comment applies equally to MSTd, 7a, and dIPFC. Yet the main results we report is one of contrast between brain areas: while 7a and dIPFC can cross-decode distance to target when retaining noise correlations, only 7a can (and not dIPFC) when eliminating noise correlations.

Sixth, we want to point out that the analysis the reviewer is highlighting here is a decoding approach; one in which we find a set of weights (or subspace, manifold) able to decode distance to target. This subspace is not able to decode other variables (all cross-validated R^2 no different from chance) and thus from the standpoint of this analysis it is inconsequential if the animal arrived at a particular distance to target via different trajectories.

We have modified the text to include reference to these new figures in the following manner:

“Three pieces of evidence suggest that the stability of the manifold coding for distance to target in dIPFC, but not that in 7a, is related to the dynamic remapping of temporally-resolved unit-to-unit couplings across densities.

*First, artificially eliminating the statistical dependencies between neurons (see Methods and **Fig. S11**) abolished the ability of dIPFC ($CV R^2 > 0$, $p = 0.84$), but not area 7a ($CV R^2 > 0$, $p = 1.17 \times 10^{-8}$), in decoding distance to target across densities (HD/LD decoding, **Fig. 3D**). Eliminating these neural co-fluctuations did not change global neural dynamics as quantified by the first two PCs, which were primarily driven by signal and not noise correlations. Further, this effect was specific to the coding of distance to target: given that a brain area was able to decode a certain task variable (e.g., 7a decoding linear velocity), eliminating the corresponding noise correlations did not abolish this area’s ability for within- or across-context decoding (see **Fig. S12A**). Moreover the impact of noise correlations on the ability to cross-decode distance to target in dIPFC was not specific to linear decoders, but also extended to a non-linear artificial neural network (**Fig. S12B**).“*

Further, we have modified the *Methods* section in order to more fully describe the shuffling control, as well as describe the non-linear ANN. The text reads:

*“To test for the role of neural co-fluctuations in allowing for distance to target decoding, we adopted a shuffling procedure that destroyed neuron-to-neuron correlations but kept signal correlations intact by matching the variable of interest in shuffled and unshuffled data (see **Fig. S11**). Namely, we first discretized distance to target in 15 bins (other discretization tested and yielded conceptually identical results). Then, since each time-point corresponded to a particular bin, for each neuron we shuffled the counts for each time point corresponding to the same bin. This procedure leaves task-variables and tuning functions unchanged (as we are shuffling like-for-like) but disrupts the correlation between neurons. We then conducted the lasso regression as detailed above. This shuffling procedure was done 10 times per neuron and session to assure reproducibility (**Fig. 3D** plots the first run, after checking results were reproduced across all shuffling runs). Further, we implemented the same analysis when matching not only the variable we were interested in decoding (i.e., distance to target), but also a second variable: linear velocity (**Fig. S11**). This further accentuates the correlation between task-variables (and thus signal correlations) in shuffled (excluding noise correlations) and unshuffled (including noise correlations) datasets (**Fig. S11C**). The main result with cross-context decoding of distance to target being impacted by the lack of noise correlations in dIPFC but not in 7a was replicated (**Fig. S11D**). Lastly, to test for specificity, we perform the same analysis on all task variables (**Fig. S12**, matching signal correlations for the variable we are decoding), and while using either linear (**Fig. S12A**) or non-linear decoders (**Fig. S12B**)”*

4) Eye movements. Previous work using this task, by some of the same authors, has shown the importance of eye movements in the performance of this task. Indeed, in that paper (ref 25) the eye movements are clearly correlated with the monkey’s sense of where it is in the virtual world, and might even carry that information explicitly. Also, ongoing and upcoming eye movements have been documented to affect neurons in all three areas under study in this paper. Eye position is included in the model, but eye velocity is not. And neither is

mentioned in the text. Given that pursuit signals in MSTd have nonlinear dynamics (Krauzlis and Lisberger), a linear position signal will probably not capture much of it. Common eye movement signals could then easily masquerade as noise correlation.

We thank the reviewer for this insightful and very important question. To address it, we have performed two new analyses. First, we compare eye-movement dynamics across density conditions. These are now presented in the supplementary materials (**Figure S3**) and show no difference across density conditions. As such, it is unclear how *common* eye movement dynamics across densities could masquerade as noise correlations that *differ* across densities.

Figure S3. Eye movements during different optic-flow density conditions. **A.** Example trial demonstrating eye movements in screen-coordinates. Upon target presentation, the monkey saccades to this location. Then, as the monkey approaches the target, this latter one progressively becomes more central and shifts downward (even if invisible at this time). The animal's gaze pursuit matches the trajectory of the target. Once the animal stops (response given), the animal saccades to a random location. **B.** Saccade rate (y-axis) as a function of time from target onset (x-axis) and optic-flow density (red = high density; black = low-density). As demonstrated by the example in **A.**, animals increase their likelihood of making saccades briefly after the target is presented. There is no difference between optic flow conditions (t-test at each time bin, all $p > 0.19$, uncorrected). Solid lines are averages across all monkeys; transparent lines are individual sessions. Error bars are ± 1 S.E.M. **C.** Correlation between the observed saccade (y-axis, top panel = horizontal axis, bottom panel = vertical axis) and the predicted saccade animals should have made if they were intending to land on the location of the target (x-axis). This is one example session, for both high- (red) and low-density (black) optic flow conditions. Dots are individual trials. **D.** Correlation coefficient (y-axis) for observed and predicted saccades along the vertical and horizontal axis, as a function of optic flow. Dots are individual sessions, error bars show the mean and ± 1 S.E.M. Saccades were better correlated with target location along the horizontal axis ($p < 0.001$), with no difference across optic-flow densities (all $p > 0.39$). **E.** Gaze pursuit for an example session. As shown by the example trial in **A.**, horizontal gaze progressively became more central, while vertically the eyes moved downward – as if tracking the invisible target. **F.** We express the correlation between the observed and predicted 2-dimensional gaze location, under the hypothesis that animals are tracking

the invisible target (see²⁵ for the original report of this effect and further methods). We call this the target-tracking index. This index was highest at target offset, and was significantly above chance (y-axis = 0) for the duration of the trials. We observed no difference between high- and low-density optic flow conditions (all $p > 0.61$, time-resolved t-test uncorrected). See REF²⁵ for the same result in different animals/sessions.

Secondly, we fit a new P-GAM encoding model, now additionally including eye movement velocity as an explicit co-variate. As the reviewer predicted, eye velocity is more readily encoded than eye position (**Figure S9B**). Most importantly to the claims at hand, as in the main text, we correlate the shape of time-resolved coupling functions estimated by this new encoding model (see **Fig. S9D**). We replicate a central finding from the manuscript (**Fig. 2D**) in showing that noise correlations more readily remap in area 7a and dIPFC than in MSTd. In fact, the coupling functions estimated via the two encoding models were very highly correlated (average $r = 0.91$) and thus also the correlation between coupling filter stability and decodability (**Fig. 3E**) and performance (**Fig. 3F**) were also maintained.

Figure S9. An alternative encoding model reproduces the dynamic remapping of noise correlations in 7a and dIPFC. **A.** Alternative encoding scheme (contrast with **Fig. 2A**, differences highlighted in orange here). This alternative P-GAM expressed distances to target in egocentric coordinates. Namely, instead of conceiving the targets in a three-dimensional space with depth, the targets are now projected onto the two-dimensional coordinates of a screen. Instead of a radial and angular distance to target, now the targets are expressed by a vertical and horizontal angle vis-à-vis the ongoing location of gaze (also on the screen). Further, we test the hypothesis that animals may not path integrate distances from origin, but from their eventual stopping locations, and thus we include egocentric distances (vertical and horizontal) to the eventual stopping location. Lastly, we include not only eye positions, but also eye velocity. **B.** Fraction of neurons tuned to the different

variables in the alternative encoding model. These are congruent with prior results (REF³⁵ and **Fig. S7A**), showing a patterned mixed selectivity, with greater coding for sensorimotor variables in 7a and of latent variables of dIPFC. Further, the results show greatest coding of eye velocity in MSTd. **C.** Signal correlations (r-value between tuning functions estimated in the different encoding models) are a function of brain area (MSTd in green, 7a in blue, and dIPFC in red) and task variable (restricted to the subset of variables present in both the main-text encoding model and the alternative one presented here). **D.** Coupling filter correlation (r-value in shape of coupling function in low- and high-density optic flow condition) as a function of brain area (MSTd in green, 7a in blue, and dIPFC in red) and encoding schema (main text on the left, and alternative P-GAM on the right). The findings replicate the results demonstrating greater remapping in 7a and dIPFC as opposed to MSTd with the change of optic-flow density. Error bars are ± 1 S.E.M.

We have modified the text in order to make reference to these new analysis:

*“Eye movements were unconstrained during the task, which allowed animals to naturally saccade to target upon its presentation (**Fig. S3A-D**). Then, the animals pursued with their gaze the evolving location of the invisible target (**Fig. S3A, E, F**), which is seemingly an innate mnemonic task strategy^{25, 35} the animals employ. There was no difference in this eye-movement behavior (saccade or pursuit) as a function of optic-flow density (see **Fig. S3** and its caption for statistical testing).”*

And:

*“This analysis showed that the temporal structure of stimuli-independent neural co-fluctuations partially remapped with context in all areas (greater than chance, smaller than max; **Fig 2D** shows full distributions as cumulative probabilities, see **Fig. S9** for a different visualization), but did so less in MSTd (average r , 0.85 ± 0.017) than in 7a (0.73 ± 0.002) and dIPFC (0.73 ± 0.014 , Kruskal-Wallis contrasting areas, $p = 0.0021$; post-hoc Holm-Sidak corrected contrasts, MSTd vs. 7a, $p = 0.003$; MSTd vs. dIPFC, $p = 0.02$; 7a vs. dIPFC, $p = 0.76$, **Fig. 2D**). To test for the generalizability of this finding, we fit an alternative encoding model wherein distances to target were expressed in egocentric coordinates, and we included distances to the eventual monkey stopping location as opposed to distance from origin. Further, we included regressors accounting for not only eye position, but also eye velocity (**Fig. S9A-C**). We confirm that also with an alternative encoding schema, areas 7a and dIPFC remapped noise correlations more readily than MSTd (**Fig. S9D**, Kruskal-Wallis, $p = 0.0023$).”*

And:

*“In addition to the main encoding model described above and in the main text, we also fit an alternative encoding model to test for generalizability of the reported effects – particularly those relating to the remapping of functional connectivity with optic-flow context in 7a and dIPFC. This alternative encoding model is defined by a different set of inputs. Namely, we include the distance to target in egocentric coordinates as opposed to a virtual three-dimensional space. Namely, we project the target onto the two dimensions of the screen, and then compute the vertical and horizontal angle between the current position of gaze, the eyes, and where the target is on screen-coordinates. We also include (in egocentric coordinates) the distances to the eventual stopping position, given that animals are tasked with stopping where they believe the target to be positioned. And thus the stopping location is a proxy for believed target position²⁷. Lastly, given that pursuit signals in MT/MST/MSTd have non-linear dynamics⁵⁰⁻⁵², we augment the model to include eye velocity components (horizontal and vertical) in addition to eye position. Differences between the encoding model used in the main text and the alternative model are illustrated in **Fig. S9A and B.**”*

References:

50. Krauzlis RJ, Lisberger SG. A model of visually-guided smooth pursuit eye movements based on behavioral observations. *J Comput Neurosci.* 1994 Dec;1(4):265-83. doi: 10.1007/BF00961876. PMID: 8792234.

51. Krauzlis RJ, Lisberger SG. Temporal properties of visual motion signals for the initiation of smooth pursuit eye movements in monkeys. *J Neurophysiol.* 1994 Jul;72(1):150-62. doi: 10.1152/jn.1994.72.1.150. PMID: 7965001.

52. Krauzlis R.J. Lisberger, S.G. A Control Systems Model of Smooth Pursuit Eye Movements with Realistic Emergent Properties. *Neural Computation*, vol. 1, no. 1, pp. 116-122, March 1989, doi: 10.1162/neco.1989.1.1.116.

5) The main interpretation of the paper rests on stability of dynamics being related to changes in coupling. They nicely document the changes (or lack thereof) in coupling, but do not show anything about the stability of dynamics or tuning across conditions. I'd like to see something akin to Fig. 2D for neuronal tuning functions, split out by the stimulus variable. Additional strength could be brought to their favored interpretation if neurons with larger numbers of coupled partners were more stable in their tuning.

We thank the reviewer for pointing out this important omission. We have included the figure requested as **Figure S7C**. As it can be appreciated, there is no change in signal correlations (beyond a change in gain in MSTd, see **Figure 2B**) across density conditions. To further demonstrate the stability of these tuning functions, we also perform signal correlations when estimating a subset of these variables via two different encoding models. This is presented in **Figure S6D** (see above), and demonstrates not only stability in the empirical tuning function of the neurons, but also in the ability of the PGAM to estimate them.

Figure S7. Signal correlations in high and low optic-flow density conditions. **A.** Fraction of neurons tuned. High- (colored) and low-density (black) optic flow conditions did not result in a different fraction of neurons tuned to different task variables in MSTd (green), area 7a (blue), or dIPFC (red). Error bars are +/- 95%CI across neurons in all sessions. **B.** Example tuning functions to linear velocity (top) and time from movement offset (bottom) for each neural area (green = MSTd; blue = area 7a; red = dIPFC) during high- (colored) and low- (black) optic flow density conditions. In addition to the raw data (solid lines), we also demonstrate the re-scaling via linear regression from low- and high-density condition (dashed lines) in order to estimate gain modulations. The examples are representative, in that they demonstrate (1) a stability in the overall shape of tuning functions across all areas (see main text), and (2) no or weak gain modulation in 7a and dIPFC, but a strong gain modulation in MSTd (see Fig. 2B). **C.** Average correlation (r-values) between the tuning functions to different task variables (x-axis) estimated in high and low optic-flow density conditions. The tuning functions are very stable (grand mean $r = 0.92$) and not different across brain areas (MSTd = green; 7a = blue; dIPFC = red). Error bars are ± 1 S.E.M across neurons (all sessions combined).

We have also checked the reviewer's prediction; the fact that neurons with a larger number of coupled partners should be more stable in their tuning. This is in fact true, as shown in as **Figure S8B**. We thank the reviewer for this insightful suggestion.

Figure S8. Noise correlations in high and low optic-flow density conditions. **A.** Fraction of units coupled within and across areas in high- (left) and low-density (right) conditions. Thickness of arrows, and inset percentage indicate the fraction of neurons coupled within and across areas. An arrow projecting from e.g., MSTd to dIPFC indicates that the firing of a neuron in MSTd will subsequently influence spiking activity in dIPFC. **B.** Signal correlation (i.e., r -value between a given tuning function estimated in low and high density) as a function of the fraction of simultaneously recorded neurons (within brain area) the unit is coupled to (top), or the total number of units (within and across area) the neuron is coupled to (regardless of the number of simultaneously recorded neurons, bottom). The units a given neuron is significantly coupled to may change across densities, and thus we perform the abovementioned computation when coupling frequency (top) and total number of coupled units (bottom) is computed either in the high (left) or low (right) optic flow density condition. In all cases, tuning stability is greater for neurons coupled to more other neurons. **C.** Coupling filters remain stable across optic flow densities in MSTd (green), but less so in area 7a (blue) and dIPFC (red). Five examples are shown for each area (high-density colored and low-density in black; shown in Kernel-space). Coupling functions within area had a length of 36ms (see¹⁶ and *Methods* for further detail). See **Figure S6E** (red) for example coupling filters in Hz-space, and protracted in time (1.2 seconds).

We must highlight that while the stability of tuning functions explains (and was predicted by) the fact that global population dynamics retain similar shapes across densities (**Figure 3A**), and the gain modulation of these tuning functions in MSTd (**Figure 2B**) explains this area's linear translation in a PCA-space, this stability in tuning functions does not account for:

- (1) the differential stability of coupling filters in MSTd vs. 7a and dIPFC,
- (2) the fact that cross-decoding is impacted by the loss of noise correlations in dIPFC but not 7a,
- (3) the correlation between remapping of noise correlations in dIPFC and ability for cross-context decoding,
- (4) the correlation between remapping of noise correlations in dIPFC and behavioral performance,
- (5) the correlation between stability of noise correlations in MSTd and behavioral performance.

The text has been modified in the following manners:

*“Regarding signal correlations, the fraction of neurons tuned to different task variables was not different across high- and low-density conditions (all $p > 0.25$, **Fig. S7A**; see³⁵ for a detailed characterization of signal correlations, including mixed selectivity^{39, 40}, during the high-density condition). Further, provided that a neuron was tuned to a given task-variable in both high and low densities (96.8% of cases), this tuning was stable in shape across densities (examples in **Fig. S7B**, grand-average correlation coefficient = 0.92 ± 0.007 , **Fig. S7C**). There was no difference in the stability of tuning function shape across densities in the different brain areas (one-way ANOVA, $p = 0.54$). Given this tuning stability, to quantify the impact of the different optic flow densities on the gain of neural tuning, we examined the gain of the linear regression between tuning functions in different densities of neurons stably tuned ($r > 0.5$, 95% of units; see **Fig. S7B** for example regressions). While we observed significant gain modulation in all brain regions (all $p < 0.005$), the effect size was considerable in MSTd (Cohen's $d = 0.21$) and negligible in the other areas (Cohen's $d < 0.06$ in 7a and dIPFC, **Fig. 2B**; examples in raw data in **Fig. S5**, and at the level of tuning functions in **Fig. S7B**). Neural responses were less prominent in MSTd (i.e., gains < 1) during the low than the high-density condition, suggesting that while signal correlations were driven by changes in environmental input in MSTd, they were less so in 7a and dIPFC.”*

And:

*“The correlation between change in tuning gain and coupling strength across sessions in MSTd showed a weak trend but was not significant ($r = -0.23$, $p = 0.19$). On the other hand, the correlation between tuning stability – as opposed to gain – showed a correlation with the number, or fraction, of neurons it coupled with (**Fig. S8B**), suggesting that unit-to-unit coupling stabilizes tuning responses across contexts.”*

And:

*“The results show that while population dynamics changed with density in MSTd, they did not in 7a and dIPFC (**Fig. 3A**, second through fourth row. See **Fig. S10** for quantification beyond the first 2 PCs and results showing that in higher-dimensions the population code in dIPFC, but not 7a, does change with optic-flow density). The change in population dynamics in MSTd appeared to be a translation between optic-flow densities, as one would*

expect given the gain-modulation of tuning functions in this area (Fig. 2B). The lack of global change in population dynamics (as indexed by PCA) in 7a and dIPFC is also in line with the lack of unit gain-modulation or remapping of signal correlations, the largest contributor to total neural variance.”

6) With regard to interpretation, the authors attach a lot of importance to the coupling, but it is relatively rare. How much influence can it have? Since we have no quantitative estimates of coupling strength (in terms of spike rates), the reader is at a loss. I think it would vastly help their conclusions if they could give the coupling strength in more interpretable units. I don't know if it is possible in their modeling, but one manipulation they should consider is to implement the model artificially, then remove all the coupling terms and observe the stability of the network in each area.

We thank the reviewer for this comment. The modeling approach undertaken attempts to estimate *when* a spike will occur (i.e., how is a neurons' output modulated by task variables), as opposed to trying to fit an overall, average firing rate. This necessitates a Poisson noise model, which renders the metrics we may derive (i.e., pseudo-R²) perhaps less common to the general audience.

Nevertheless, as suggested by the reviewer we can indeed fit encoding models with and without coupling filters. The average pseudo-R² (a metric appropriate for Poisson noise) is 0.091 under the coupled model, and 0.063 under the uncoupled model. In other words, the fits allowing for coupling between neurons are 1.5 times better than those without coupling (**Figure S6B**). We can also produce spike trains from the coupled and uncoupled model, convolve these spike trains, and compare root mean squared errors (although we must note the exact values will depend on the convolution undertaken). **Figure S6C** shows snippets of real data, as well as firing rates produced by coupled and uncoupled models. The coupled model produces an R² that is 1.14 times better than the uncoupled model. Altogether, the coupling does allow accounting for more neural variance. More important, however, is that it specifically can account for neuron-to-neuron correlations, which are completely unaccounted for in the uncoupled model (see **Figure S6E** and associated figure caption).

Lastly, in order to provide a measure in terms of spike rates, as suggested by the reviewer, we compute the mean evoked (i.e., peak response above or below firing rate) spike-triggered average, as estimated by the model. Importantly, we only do this for neurons deemed to be significantly coupled according to the P-GAM, given that the “raw” approach vastly over-estimates the likelihood of neurons being coupled (as this approach does not take into account responses that are driven by signal, see caption of **Figure S6E**). The resulting values on the order of ~1 spike/s are in line with empirical evidence estimating a modest impact of a neuron's firing on spiking activity of a coupled neuron (Chettih & Harvey, 2019, Nature).

We have modified the text in the following manner:

*“The P-GAM accounted well for spiking activity (average pseudo-r² = 0.091, ~1.5 to 2 times better than traditional Generalized Linear Models³⁸, see **Fig. S6A** for example tuning functions, both as raw binned responses and as estimated by the P-GAM), and did so better when including (vs. excluding) coupling filters (~1.5 times better in pseudo-r² and ~ 1.14 times in R², **Fig. S6B** and **C**). Importantly, the addition of coupling filters did not change the estimate of tuning functions (correlations between tuning functions estimated with and without coupling functions, $r \sim 0.93$, **Fig. S6D**), but critically, did allow to additionally account for signal-independent inter-neuron correlations (model prediction vs. empirical spike-triggered average, coupled model, $R^2 = 0.65 \pm 0.02$; uncoupled model, $R^2 = 0.26 \pm 0.06$; $p = 0.0003$; see **Fig. S6E** for examples and caption for detail).”*

And:

*“The coupling strength (estimated as area under the coupling filter in kernel-space, collapsed across time) did not change with optic flow density within 7a or dIPFC (both $p > 0.67$, both Cohen's $d < 0.05$). On the other hand, coupling strength within MSTd increased during the low-density manipulation ($p < 0.005$, paired t-test, Cohen's $d = 0.51$; **Fig. 2C**). The evoked (i.e., peak response above or below baseline firing) spike-triggered average (in Hz-space, **Fig. S6E**) in coupled neurons in MSTd during the high-density condition was 0.64 Hz (± 0.09 Hz), and 1.02 Hz (± 0.11 Hz) in the low-density condition ($p = 0.011$). On average, inter-neuron spike-triggered averages were 1.13 Hz and 1.08 Hz, respectively in 7a and dIPFC (no difference across densities, both $p > 0.36$).”*

And:

“Pseudo- R^2 . We computed pseudo- R^2 s to assess fit quality. This metric is a goodness of fit measure that is suitable for models with Poisson observation noise⁵³, and is computed as,

$$\text{pseudo } R^2 = 1 - \frac{\mathcal{L}(y) - \mathcal{L}(\hat{y})}{\mathcal{L}(y) - \mathcal{L}(\bar{y})} \quad (\text{Eq. 4})$$

with $\mathcal{L}(y)$ being the likelihood of the true spike counts, $\mathcal{L}(\hat{y})$ being the likelihood of the P-GAM model prediction, and $\mathcal{L}(\bar{y})$ being the likelihood of a Poisson null-model (constant mean rate). We computed this measure on held-out test trials (20% of the total trials, not used for inferring model parameters). The pseudo- R^2 can be interpreted as the fraction of the maximum possible likelihood (normalized by the null model) that the fit attains. The score is 0 when the P-GAM fits are no more likely than the null model (constant mean rate), 1 when it perfectly matches the data, and can be negative when overfitting occurs. In this latter case, we excluded the unit from analysis (2.7% of the recorded units). Empirically, the pseudo- R^2 is a stringent metric⁵⁴ and ranges in values that are substantially lower than the standard R^2 (see **Fig. S6B** for a comparison of pseudo- R^2 in coupled and uncoupled P-GAMS). We also provide an R^2 metric (average = 0.26, no different across densities, $p = 0.91$) as a comparison to prior studies (current study is better than its most similar report⁴⁴ with a reported average of 0.15) and for ease of interpretability, but it must be noted that this latter metric relies on first convolving spike trains (here we used a Gaussian filter with standard deviation = 20ms, **Fig. S6C**) and will change depending on the convolution filters used.

Coupling Filters. Coupling filters (and the corresponding inferential statistics) were determined via the P-GAM³⁰. Within area coupling filters were set to a duration of 36ms, and across area filters were initially set to a duration of 600ms. We focus our analyses *in the main text* exclusively on the within area coupling filters, given their higher presence rate, and the fact that we can be more confident in indexing unit-to-unit interactions when restricting the analyses to a short time-frame (i.e., 36ms). *For further scrutiny of the coupling filters and their contrast to empirically-derived cross-neuron spike-triggered averages (Fig. S6E) we additionally fit encoding models with longer coupling (1.2 seconds), or without coupling filters altogether (e.g., Fig. S6D and E).*”

7) The coupling filters are very short, and all the examples show them to be highly asymmetrical. This does not resemble the coupling between neurons measured with more traditional methods, which is often rather extended in time.

We thank the reviewer for this question. Our coupling functions are “causal” or “directional” (from sender unit to receiver unit) and thus there is no notion of symmetry or asymmetry. Further, the coupling functions were purposefully kept very short in an attempt to index true unit functional connectivity as opposed to another non-specific effect (e.g., volume conductance). However, we appreciate the need to demonstrate that our method and more traditional approaches yield similar results. Thus, we have (1) fit encoding models with extended coupling filters (1.2 seconds as opposed to 36ms), and (2) show the overlap (see examples in **Figure S6E**) between empirically derived coupling functions and those estimated via “brute” spike-trigger averaging. The mean R^2 of the coupled model accounting for inter-neuron correlations is 0.75 (± 0.02 , assuming both the P-GAM and “brute” approach agreed on indexing a pair as coupled).

Figure S6. Coupling filters account for inter-neuron correlations. **A.** Example tuning, spike-history, and coupling filters estimated by the P-GAM. From top to bottom: Raw (gray) and P-GAM reconstructed (black) filters for task-variables (demonstrating the model's ability to account for spiking activity), spike history filters (showing the characteristic refractory period of single units), and coupling. **B.** Pseudo- R^2 for uncoupled (x-axis) and coupled (y-axis) models, showing that P-GAM encoding models allowing for neuron-to-neuron coupling accounted better for observed spike trains. Each dot is a neuron and the identity line in red (note different range in x and y). **C.** Five snippets (each 30 seconds long) of "observed" neural activity (convolved with a Gaussian filter with a standard deviation of 20ms, black), as well as predicted firing rates with a P-GAM allowing for coupling (coupled model, red), or not (uncoupled mode, blue). **D.** Correlation between tuning functions to different task variables (x-axis) estimated with the coupled and uncoupled model (y-axis = r-values, average $r = 0.93$, all variables above $r = 0.85$). The different brain areas are colored (MSTd = green; 7a = blue, dlPFC = red), and there was no difference in the stability of these tuning functions across brain area (one-way ANOVA, $p = 0.68$). Note, however, there was a difference in their gain-modulation, being most readily modulated in MSTd than 7a or dlPFC (see **Fig. 2B**). **E.** Example (12) across-units spike-triggered averages (i.e., average firing rate of "receiver" neuron condition on a spike from a "sender" unit), as empirically estimated (black) and estimated via coupled (red) or uncoupled (blue) P-GAM encoding models. Ten of these neurons (all but bottom row, 3rd and 4th from the left; transparent colors) were estimated as coupled by the P-GAM, while the last two (shown as a negative control) were determined not to be coupled. Of note, the coupled model (red) is able to recapitulate neuron-to-neuron dynamics, while the uncoupled model is not (mean R^2 , coupled model, 0.65 ± 0.02 ; uncoupled model, $R^2 = 0.26 \pm 0.06$; $p = 0.0003$). Further, we determine 95% CIs for each sender-receiver pair by shuffling spike times of the "sender" unit 1000 times and re-computing spike-triggered averages. On average, the P-GAM estimated 11.31% of unit pairs to be coupled, while this number was much higher (39.8%) according to permutation testing on raw data. This discrepancy highlights the critical need for an encoding model, accounting and explaining away signal-correlations. In other words, while the empirical spike-triggered averages may be driven by both neurons responding to a "signal", the model estimated pairs are conditioned on this other input, and thus results in a more conservative (i.e., statistically robust) estimate. When both the coupled model and empirically-estimated spike triggered averages agreed on indexing a coupled pair, their R^2 was 0.75. In conclusion, the real benefit to the coupled P-GAM is not in better accounting for spike times of a single neuron (**B** and **C**) but in being able to account for neuron-to-neuron dynamics above and beyond signal correlations (**E**, also see **Figure 4C** in Lakshminarasimhan et al., 2023).

We must mention 2 important caveats. First, the P-GAM and raw coupling functions are not quite indexing the same phenomenon. The "brute approach" is incapable of indexing noise correlations that are independent of signal correlations (all activity is considered). Instead, the "P-GAM approach" is signal-independent, at least in as much as we are able to account for "signal" given our inputs. We hypothesize that the correlation between coupling functions estimated via P-GAM and "raw" approaches would increase if we limited this analysis to neurons not responding to any other variable, but no neuron met this criteria (for example, a very large majority is tuned to ongoing phase of LFP). Second, we must mention that here our goal is to use an encoding model to attempt to predict the timing of spiking activity. Thus, including "symmetrical" coupling functions in non-sensical, as activity of unit X *following* a spike from unit Y cannot predict activity of unit Y.

Smaller concerns:

8) In their description of the task, they say that the texture elements can't be used as landmarks. This is technically true, but is a little disingenuous, since the monkeys have several cues available to be used for landmark-based navigation. These include texture element size and density, and target location on the screen.

The target briefly flashes and then disappears for the duration of the trial. The texture elements are re-drawn, re-positioned, and re-oriented randomly every 250ms. Given these characteristics those objects will clearly provide a sense of perspective (i.e., depth) and velocity given their size and density. However, we fail to see how they can be used to triangulate the dynamically evolving location of the hidden target (e.g., "target is X cm left from landmark"). In any case, we delete from the revised manuscript mention to "landmarks" given it is not central to the conclusions and in case this is confusing or appears disingenuous – not our intention.

9) The methods need to be much better described. There is no mention of number of trials or spikes for each of the neurons, nor of the number of neurons and how repeated neurons across sessions were treated. More descriptions of the assumptions and limitations of the fitting procedure also need to be given for the non-specialist audience.

We have now included in the *Methods* section the average number of trials per session and the number spikes per neuron (i.e., firing rate). The number of neurons per area was included in the original submission. The text has been modified as follows:

“PR is 1 minus the fraction of 1-minute bins in which there is no spike. We set the following thresholds in qualifying a unit as a single-unit: ISlv < 20%, cR < 0.02, and PR > 90%. After spike sorting, the final neural dataset consisted of 3332 single units (MSTd: 240; parietal area 7a: 2647; dIPFC, 445 neurons), with an average of 40 units simultaneously recorded per session. The average firing rate of units in MSTd, 7a, and dIPFC were, respectively, 5.69, 6.14, and 6.61 spikes/s.”

And:

“Monkeys typically performed blocks of 500-750 trials before being given a short break. In a session, monkeys would perform 2 or 3 blocks. High (5.0 elements/m²) and low (0.1 elements/m²) density conditions were randomly intermixed with equal probability within each block. On average, animals attempted 1070.26 trials per session, resulting in a total dataset of 44476 trials in the high-density condition and 43286 in the low-density condition.”

Neurons were treated as independent units across sessions (a common approach with Utah array recordings). We recorded in all three areas (MSTd, 7a, and dIPFC) with linear probes that were lowered into the brain each day, and in different positions. In addition to these “acute” preparations – where we can be confident we are not recording from the same unit on different days – we also performed “chronic” recordings with Utah arrays in dIPFC and 7a. These Utah array recordings allow for a higher yield, yet are not possible in MSTd given its anatomical location (deep in the sulcus). In order to ascertain whether the assumption of independence was warranted (or at least did not bias the effects we report), we checked whether the differential noise correlations reported in MSTd, 7a, and dIPFC were true across recording techniques (i.e., linear probe and Utah array). As shown in **Figure S14**, noise correlations remapped with density condition in 7a and dIPFC, and less so in MSTd, independently of the recording technique used. There was no difference in coupling filter correlation as a function of recording technique in 7a ($p = 0.71$) nor dIPFC ($p = 0.79$).

Figure S14. Coupling filter correlation as a function of probe type. MSTd recordings were undertaken exclusively with acute recordings, due to its anatomical location. Recordings in 7a and dIPFC were undertaken both with acute and chronic preparations. To examine if probe type impacted the results (and control for the possibility that units recorded on different days with chronic preparations are not independent), we examined coupling function correlations as a function of probe type. Even when restricting our analyses to acute recordings, noise correlations in 7a and dIPFC more readily remapped as a function of optic-flow context (Kruskal-Wallis, $p = 0.0035$). There was no difference in coupling filter correlation as a function of recording technique in 7a ($p = 0.71$) nor dIPFC ($p = 0.79$).

We have amended the text in the following manner:

“Chronic setups were used for their higher yield, while acute linear probe recordings were used in order to target MSTd, which is deep. Further, we used both chronic and acute preparations to ascertain that neural results were independent of the type of probe used (Fig. S14). During acute recordings with the linear arrays, the probes were advanced into the cortex through a guide-tube using a hydraulic micro-drive.”

10) p. 6, para 3. Description of gain analysis is confusing, and might be wrong. The legend to Figure S3C talks about using a scale factor to estimate a gain change, which could be correct, but the main text confusingly talks about using “the slope of this correlation” to make the same estimate. Surely there’s a more straightforward way to estimate gains, or to describe their approach.

We apologize for the confusion. We have modified the text for clarity. The text reads:

*“A linear model with multiplicative gain (i.e., $\text{response} = \text{gain} \times \text{target location} + \text{intercept}$; see^{25-28, 37} for the same approach) accounted well for the observed data (average $R^2 = 0.72$). Thus, we used the gain term of the corresponding linear regressions as a measure of behavioral performance (gain = 1 would indicate no over- or under-shooting of targets, gains < 1 indicate undershooting, and gains > 1 indicate overshooting. The average intercept = 1.1cm in radial distance and 0.2 deg, neither significantly different from 0, both $p > 0.81$; **Fig. 1B and C**).”*

11) p. 6, last paragraph. The phrase “...as if attempting to offset the unavailability...” makes little sense to me, and the costs and benefits of coupling depend on details of signal and noise that are unavailable. In any case, it is highly speculative and doesn’t belong in the results section.

We agree with the reviewer, this sentence has been eliminated.

Reviewer #2 (Remarks to the Author):

In this manuscript, Noel et al study the neural mechanism underlying context invariant computations, i.e., the ability of animals to generate similar behaviors in settings characterized by different sensory inputs. The authors record neural population responses from visual, parietal, and prefrontal areas in monkeys engaged in a task requiring them to navigate with a joystick towards a remembered target location. The monkeys perform this task in two contexts, one in which they can infer their position in the environment based on strong visual cues, and one where the visual cues are impoverished, thus forcing the monkeys to rely more on learned internal models capturing the relation of joystick inputs and the resulting movement within the environment.

The authors analyze population responses with two complementary approaches that are typically not considered at the same time: (1) a new framework they call generalized additive model, similar to GLM, in particular to infer the nature of coupling between neurons that is responsible for “noise correlations”; (2) low-dimensional population trajectories and decoding. These approaches reveal several potentially interesting features of population responses in the recorded areas. One main finding is that context-invariant dynamics (at the level of trajectories and decoding) seems to correlate with a context-dependent “remapping” of the coupling between neurons. This relation is evident in a comparison across areas, as well as across sessions in PFC.

I find the general question and methods timely and important. The results are intriguing and potentially of broad relevance. However, the authors should perform additional analyses and controls to help in the interpretability of their findings, and/or to establish a more direct relation to past models of context-dependent computations.

Main comments:

(1) In the description of the behavior, I did not find any explanation of the eye movements of the monkeys during the task. Are monkeys fixating? Or are they free to move the eyes? Based on their REF 35, I assume the eyes are free to move. If so, it seems imperative to compare eye movements across contexts, as such movements could have a strong impact on neural responses in the recorded areas.

We thank the reviewer for this comment and agree that comparing eye movements across contexts is an imperative for this manuscript (also noted by R1). The animals were indeed free to move their eyes as they will. We have added a supplementary figure (**Figure S3**) where we examine (1) saccade frequency and timing, (2) saccade directions and magnitude vis-à-vis the prediction that animals will saccade to target, and (3) gaze pursuit under the hypothesis that animals follow the (invisible) target with their eyes. The results replicate our prior findings (e.g., REF^{25, 35}) indicating that the animals saccade toward target and then pursue it for the duration of the trial. Most importantly for the current purposes, we observed no difference across optic-flow densities (see Figure and figure caption below).

The main text has been modified in the following manner:

*“Eye movements were unconstrained during the task, which allowed animals to naturally saccade to target upon its presentation (**Fig. S3A-D**). Then, the animals pursued with their gaze the evolving location of the invisible target (**Fig. S3A, E, F**), which is seemingly an innate mnemonic task strategy^{25, 35} the animals employ. There was no difference in this eye-movement behavior (saccade or pursuit) as a function of optic-flow density (see **Fig. S3** and its caption for statistical testing).”*

The following figure and caption have been added to the supplementary materials:

Figure S3. Eye movements during different optic-flow density conditions. **A.** Example trial demonstrating eye movements in screen-coordinates. Upon target presentation, the monkey saccades to this location. Then, as the monkey approaches the target, this latter one progressively becomes more central and shifts downward (even if invisible at this time). The animal's gaze pursuit matches the trajectory of the target. Once the animal stops (response given), the animal saccades to a random location. **B.** Saccade rate (y-axis) as a function of time from target onset (x-axis) and optic-flow density (red = high density; black = low-density). As demonstrated by the example in **A.**, animals increase their likelihood of making saccades briefly after the target is presented. There is no difference between optic flow conditions (t-test at each time bin, all $p > 0.19$, uncorrected). Solid lines are averages across all monkeys; transparent lines are individual sessions. Error bars are ± 1 S.E.M **C.** Correlation between the observed saccade (y-axis, top panel = horizontal axis, bottom panel = vertical axis) and the predicted saccade animals should have made if they were intending to land on the location of the target (x-axis). This is one example session, for both high- (red) and low-density (black) optic flow conditions. Dots are individual trials. **D.** Correlation coefficient (y-axis) for observed and predicted saccades along the vertical and horizontal axis, as a function of optic flow. Dots are individual sessions, error bars show the mean and ± 1 S.E.M. Saccades were better correlated with target location along the horizontal axis ($p < 0.001$), with no difference across optic-flow densities (all $p > 0.39$). **E.** Gaze pursuit for an example session. As shown by the example trial in **A.** horizontal gaze progressively became more central, while vertically the eyes moved downward – as if tracking the invisible target. **F.** We express the correlation between the observed and predicted 2-dimensional gaze location, under the hypothesis that animals are tracking the invisible target (see²⁵ for the original report of this effect and further methods). We call this the target-tracking index. This index was highest at target offset, and was significantly above chance (y-axis = 0) for the duration of the trials. We observed no difference between high- and low-density optic flow conditions (all $p > 0.61$, time-resolved t-test uncorrected). See REF²⁵ for the same result in different animals/sessions.

In addition, we have fit an alternative encoding model explicitly accounting not only for eye position, but also eye velocity. This new encoding model recapitulates the main conclusion of the manuscript, that noise correlations change with optic-flow density most readily in area 7a and dIPFC than MSTd. Further, the correlation between coupling filters estimated with each encoding model is $r = 0.91$. The findings from **Figure 3** in the main text are replicated. The new encoding model is included in the supplementary as:

Figure S9. An alternative encoding model reproduces the dynamic remapping of noise correlations in 7a and dIPFC. **A.** Alternative encoding scheme (contrast with Fig. 2A, differences highlighted in orange here). This alternative P-GAM expressed distances to target in egocentric coordinates. Namely, instead of conceiving the targets in a three-dimensional space with depth, the targets are now projected onto the two-dimensional coordinates of a screen. Instead of a radial and angular distance to target, now the targets are expressed by a vertical and horizontal angle vis-à-vis the ongoing location of gaze (also on the screen). Further, we test the hypothesis that animals may not path integrate distances from origin, but from their eventual stopping locations, and thus we include egocentric distances (vertical and horizontal) to the eventual stopping location. Lastly, we include not only eye positions, but also eye velocity. **B.** Fraction of neurons tuned to the different variables in the alternative encoding model. These are congruent with prior results (REF³⁵ and Fig. S7A), showing a patterned mixed selectivity, with greater coding for sensorimotor variables in 7a and of latent variables of dIPFC. Further, the results show greatest coding of eye velocity in MSTd. **C.** Signal correlations (r-value between tuning functions estimated in the different encoding models) are a function of brain area (MSTd in green, 7a in blue, and dIPFC in red) and task variable (restricted to the subset of variables present in both the main-text encoding model and the alternative one presented here). **D.** Coupling filter correlation (r-value in shape of coupling function in low- and high-density optic flow condition) as a function of brain area (MSTd in green, 7a in blue, and dIPFC in red) and encoding schema (main text on the left, and alternative P-GAM on the right). The findings replicate the results demonstrating greater remapping in 7a and dIPFC as opposed to MSTd with the change of optic-flow density. Error bars are ± 1 S.E.M.

The main text has been modified as:

“To test for the generalizability of this finding, we fit an alternative encoding model wherein distances to target were expressed in egocentric coordinates (see Methods), and we included distances to the eventual monkey

stopping location as opposed to distance from origin. Further, we included regressors accounting for not only eye position, but also eye velocity (**Fig. S9A-C**). We confirm that also with an alternative encoding schema, areas 7a and dIPFC remapped noise correlations more readily than MSTd (**Fig. S9D**, Kruskal-Wallis, $p = 0.0023$).”

And in the *Methods* section we add:

“In addition to the main encoding model described above and in the main text, we also fit an alternative encoding model to test for generalizability of the reported effects – particularly those relating to the remapping of functional connectivity with optic-flow context in 7a and dIPFC. This alternative encoding model is defined by a different set of inputs. Namely, we include the distance to target in egocentric coordinates as opposed to a virtual three-dimensional space. Namely, we project the target onto the two dimensions of the screen, and then compute the vertical and horizontal angle between the current position of gaze, the eyes, and where the target is on screen-coordinates. We also include (in egocentric coordinates) the distances to the eventual stopping position, given that animals are tasked with stopping where they believe the target to be positioned. And thus the stopping location is a proxy for believed target position²⁷. Lastly, given that pursuit signals in MT/MST/MSTd have non-linear dynamics⁵⁰⁻⁵², we augment the model to include eye velocity components (horizontal and vertical) in addition to eye position. Differences between the encoding model used in the main text and the alternative model are illustrated in **Fig. S9A and B**.”

(2) Several features of the GAM are not explicitly justified in this manuscript (although they may have been addressed in previous papers by the authors employing similar methods), and it remains unclear how somewhat different formulations of this model would change the results. Addressing the points below seems important to interpret the reported findings.

We thank the reviewer for this comment. We address each of the points below.

(2a) The model includes a number of external and latent variables, but some prominent variables could arguably have been added. For example, variables reflecting sensory or latent variables in retinotopic coordinates; a variable encoding context (strong or weak optic flow); a variable encoding the passage of time in a trial (which appears to be encoded e.g. in PFC, based on the data in this paper and based on past research); or variables encoding eye movements beyond the horizontal and vertical gaze location (for example eye velocity, which would encode saccades and could lead to strongly correlated firing across neurons). How would the results change if these variables were included?

We thank the reviewer for this comment. We have built a new encoding model, now expressing sensory and latent variables in egocentric or “retinotopic” coordinates and including eye velocity in addition to eye position (**Fig. S9**, see reply to Question #1). The results (both in terms of noise and signal correlations) were unchanged. The average correlation between coupling filters estimated in each of the encoding models was 0.91 (and thus why none of the results regarding this variable changed), and the average correlation between tuning functions estimated in each of the encoding models (for the subset present in both) was 0.92.

(2b) The GAM includes both filters implementing tuning for particular variables (like distance to target) as well as temporal filters (spike history and coupling). But it seems that the tuning filters are not directly linked to a “temporal” filter. Does this mean that these variables have instantaneous effects on neural responses? In many areas, the onset of a stimulus for example leads to a somewhat transient activation. How would such a temporal effect be captured in this model?

The model has two types of “signal correlations” (i.e., elements placed on the left of **Figure 2A**). A number of task variables will indeed evoke a transient activation, which could occur at different delays in different neurons or brain areas. As noted by the reviewer, this may include the onset of the stimulus. It may also include the onset and offset of movement and reward delivery. The timing of these variables were included in the model and thus we estimated a temporal filter, from 500ms prior to 500ms post-stimulus or event onset. Examples of these tuning functions (where the x-axis is time, as opposed to variable value) are included in **Figure S7B** (bottom row).

We have modified the text to more explicitly indicate which variables are included in the model as temporal filters. The text reads:

“The resultant tuning functions for continuous covariates (x_t) will have as x-axis values along the range of the particular variable (e.g., cm, cm/s), while the tuning functions for discrete events (z_t ; time of target onset, movement onset and offset, and time of reward) will have time (e.g., seconds) as x-axis. A unit’s log-firing rate was modeled as a linear combination of arbitrary non-linear functions of the covariates,

$$\log \mu = \sum_j f_j(x_j) + \sum_k f_k * z_k \quad (\text{Eq. 1})$$

*where * is the convolution operator, and the unit spike counts are generated as Poisson random variables with rate specified by (Eq. 1).”*

(2c) In this paper, the authors do not show any evidence that their method can retrieve parameters of a ground truth model (“model recovery”). The authors also do not explore cases of “model mismatch”, where the structure of the fitted model does not match that of the ground truth. For example, due to failure to include some input variables that are present in the ground truth, but are not included in the fit; or inclusion of variables that are correlated with the ground truth variables, but not identical to them. Generally, it would be comforting to see that the results are somewhat robust to such mismatch.

Please see the extensive answer to Question 1 by Reviewer 1.

(3) The authors report several potentially interesting relations between insights from their GAM model (e.g. how neuron couplings change across contexts) and population trajectories/decoding (e.g. whether representations are context-invariant or not). These relations are interpreted in terms of past models of context-dependent computations, but some of the related conclusion would be stronger with some additional analyses.

We thank the reviewer for this comment. We address each of the points below.

(3a) The authors compare task representations and their context-invariance at the level of population trajectories. But could one also employ the GAM stimulus filters? For example, does invariance of average trajectories imply that the stimulus filters should also be invariant? Did the authors evaluate the similarity of stimulus filters across contexts in the same way as they did for coupling filters?

This is a great question, and was also posed by R1. Please see our extensive answer to Question 5 by Reviewer 1.

(3b) Artificially eliminating correlations results in distance to target not being decodable anymore across contexts in PFC. I found this result surprising, as the trajectories in Fig. 3A seem to show a component of the dynamics that is (1) invariant across context; (2) encodes distance to target. Distance appears to be encoded non-linearly, as a combination of distance from the “origin” and angle within the 2d-plane (the rotation from the bottom right to the top left). Would a different, non-linear decoding approach (e.g. an artificial neural network) result in good decoding even after removing correlations?

We thank the reviewer for this great question. To address it, we built a non-linear artificial neural network for decoding. The results are presented in **Figure S12**, showing no conceptual difference between linear and non-linear decoders (while on average the non-linear decoder does perform slightly better across all task variables).

In the revised manuscript we have included the new **Figure S12**:

A - linear decoder

B - non-linear decoder (ANN)

Figure S12. Decoding task variables within and across contexts, including and excluding noise correlations. A. Decoding cross-validated R^2 for each continuous task variable (panels) and brain area (MSTd = green, 7a = blue, dIPFC = red), both within contexts (HD/HD, LD/LD) and across contexts (HD/LD). Squared filled markers are used to indicate decoding including noise correlations (i.e., “unshuffled”) while empty circles are used to indicate decoding excluding noise correlations. The latter are only plotted when the decoding for a specific area/task variable is significant when including task variables. Error bars are $\pm 95\%$ CI, and thus non-overlapping error bars with CV = 0 (dashed line) is significant. The only brain area/task variable that is decodable across context with noise correlations but not without is distance to target in dIPFC. **B.** Same as **A**, while employing a non-linear artificial neural network for decoding. Results are replicated.

And in the main text:

“First, artificially eliminating the statistical dependencies between neurons (see Methods and Fig. S11) abolished the ability of dIPFC ($CV R^2 > 0$, $p = 0.84$), but not area 7a ($CV R^2 > 0$, $p = 1.17 \times 10^{-8}$), in decoding distance to target across densities (HD/LD decoding, Fig. 3D). Eliminating these neural co-fluctuations did not change global neural dynamics as quantified by the first two PCs, which were primarily driven by signal and not noise correlations. Further, this effect was specific to the coding of distance to target: given that a brain area was able to decode a certain task variable (e.g., 7a decoding linear velocity), eliminating the corresponding noise correlations did not abolish this area’s ability for within- or across-context decoding (see Fig. S12A). Moreover, the impact of noise correlations on the ability to cross-decode distance to target in dIPFC was not specific to linear decoders, but also extended to a non-linear artificial neural network (Fig. S12B).”

And Methods:

“Decoding, Non-linear Artificial Neural Network (ANN). To test whether the reported impact of noise correlations on the ability of dIPFC to cross-decode the key latent task-variable is specific or not to linear decoding, we build a 2-layer ANN to non-linearly decode continuous task variables (e.g., distance to target, linear velocity, etc.). Data pre-processing was akin to that for the Lasso regression (above), and we used sessions with at least 10 units simultaneously recorded in a given area. The first layer of the ANN (input) had 10 neurons, with the second layer being an expansive layer with 20 neurons. There was an output layer of a single neurons. The input and hidden layers had ReLu activation functions, and we used a Mean Squared Error (MSE) loss function with L2 regularization to prevent overfitting. The training was performed using the Adam optimizer from the Optax library⁵⁵ with adaptive learning rates. Weights were initialized using a normal distribution and with no bias. Overall, the non-linear decoder performed better than the Lasso regression (mean $CV R^2$ across all variables, within context: regression = 0.171 ± 0.012 ; ANN = 0.196 ± 0.015 , $p = 0.019$; across-context: regression = 0.155 ± 0.015 ; ANN = 0.169 ± 0.018 , $p = 0.023$). The specific loss of the ability to decode distance to target across optic-flow contexts in dIPFC remained (Fig. S12B).”

We do want to highlight that **Figure 3A** are PCA projections for all neural activity, as opposed to solely distance to target. It is, hence, hard to visually interpret what these trajectories mean for population-level encoding or decoding of distance to target. Trajectories are also longer in time for far rather than near targets, even if time-warped for eventual averaging.

(3c) The interesting relation in Fig. 3E (dIPFC) is shown only for distance to target. I understand the important of that variable, but according to the GAM many other variables are also represented, and one could equally ask if their representation become more task invariant as a function of the remapping of coupling filters. Does decoding of those variables show a similar relation?

This is an excellent question. We address it in 2 ways. First, we trained decoders and test for cross-context decoding of all variables. Given that an area is capable of decoding a particular task variable across contexts, we (1) study the impact of artificially eliminating noise correlations via a shuffling procedure akin to that performed in the main text, and (2) more directly addressing the reviewers question, we examine the correlation between cross-context decoding and coupling stability.

Figure S12 (see question above) shows the cross-validated R^2 when decoding different task variables both within (HD/HD or LD/LD) and across (HD/LD) optic-flow density contexts. All areas are shown: MSTd in green, 7a in blue, and dIPFC in red. Further, given that the decoder leaving noise correlations intact (colored squared marker) was better than chance (error bars are $\pm 95\%CIs$, and thus no overlap between error bars and $CV R^2 = 0$ signifies the decoder was better than chance), we also examined decoders in which the noise correlations were eliminated (open circular marker). As it can be seen in **Figure S12**, the only variable in which eliminating noise correlations rendered the cross-context decoding no different from chance was for distance-to-target (top rightmost panel). This suggests that coupling filters most readily assist in the stability of population codes for distance to target in dIPFC.

More directly addressing the question posed by the reviewer, we correlate the ability of the different areas for cross-context decoding of a particular task variable and coupling filter stability. The results are presented in **Figure S13**. The only areas/task variables pairs that showed a correlation between coupling stability (within a session) and cross-context decoding were: (1) dIPFC and the cross-decoding of distance to target (as mentioned in the main text in the original submission), and (2) 7a and the cross-decoding of linear acceleration.

We have included in the revised manuscript **Figure S13**:

Figure S13. Relations between cross-context decoding and coupling filter stability. All continuous variables are showing for dIPFC (red). The only significant correlation is between the stability to coupling filters (within a session), and the ability of that session for cross-context decoding of distance to target (top row, second column). The only significant relationship for area 7a is between coupling stability and linear acceleration (bottom row, third column).

Further, we have included the following text:

*“Second, there were strong and specific session-to-session correlations between the stability of coupling filters in dIPFC and the ability to decode **the key task variable** across contexts. That is, the more unit-to-unit couplings remapped within dIPFC in a session, the better was this session’s ability to decode distance to target across density contexts ($r = -0.71$, $p = 0.0018$; **Fig. 3E**). This was not true in dIPFC for any other task variable (**Fig. S13**), nor was it for area 7a, where coupling filter stability did not correlate with decoding generalizability for distance to target ($r = 0.08$, $p = 0.65$). In fact, where we did observe a correlation between the ability for cross-context decoding and stability of coupling functions in 7a was for linear acceleration ($r = 0.59$, $p = 7.1 \times 10^{-5}$, **Fig. S13**). The stability of tuning functions across densities did not correlate with decoding generalizability of distance to target in 7a ($p = 0.21$) or dIPFC ($p = 0.12$).”*

And:

“Fittingly, we observe that higher PCA dimensions (i.e., accounting for less variance and thus less driven by task inputs and more by noise correlations) varied with context in dIPFC but not in 7a (Fig. S10), and that coupling stability related to the ability to cross-decode linear acceleration (Fig. S13).”

“(3d) Could the authors explore the relation of stimulus filter, coupling filters, and dynamics in previously proposed recurrent neural network models of context-dependent computations? Analyzing such RNN would probably require some changes in the definition of the GAN (as the RNN units are not spiking) but potentially an account of variability in terms of stimulus filters and coupling filters would nonetheless appear meaningful. If such an analysis was possible, it would help in establishing a connection to the past work that the authors use to frame their results.”

We thank the reviewer for this question. We agree that extending this analysis to RNNs would certainly be interesting in future work. However, at the moment we consider it pre-mature for a number of reasons:

- (1) As mentioned by the reviewer, there are a great number of differences between the data we recorded here and standard RNNs (e.g., spiking vs. non-spiking). It is unclear what a presence or lack of correspondence would teach us.
- (2) Our group is currently developing RNNs to perform the navigate-to-target task we used here. In this work (Zhang et al., 2022, preprint) we have observed that the details of how the model is implemented (e.g., RNN vs. RNN+MLP; modular vs. holistic) and trained will greatly impact the ability of the model to generalize and what type of errors it makes. We consider it is most prudent to first develop an RNN to the best of our ability, and then check for agreement with biology. At the moment we could test our current results with a zoo of potential models, but this approach lacks rigor.
- (3) Our current RNNs (Zhang et al., 2022) are not image-computable and there are no eyes, and thus no eye movements. In prior work we have demonstrated how important eye movements are to the correct execution of this task (Lakshminarasimhan et al., 2020) and to neural coding within it (Noel et al., 2022). Thus, we consider we should first include eye movements and develop image-computable RNNs (such that the inputs are the same in biological and artificial networks). Developing this next version of RNNs is a project in itself.

We do, however, agree that extending our discussion to include further contextualization of our results in light of prior RNN work is warranted. We have amended the text in the following manner:

“Mechanistically, some⁶ have argued that a change in context results in an alteration in recurrent dynamics. Others⁷ have assumed that connectivity does not change at this time scale, and thus neither can the underlying neural dynamics. Instead, according to this latter view, changes in context ought to alter the initial condition or input into the dynamical system (also see REF⁴⁶). At first glance, it would appear that our results support the former interpretation, given that we report changes in functional connectivity. However, we must highlight that given the closed-loop nature of our task^{25-28, 47}, input to the system are state-dependent and continuously changing. This renders challenging, if not impossible, the teasing apart of the relative contributions of internally generated dynamics from external inputs⁴⁸. In future work, this arbitration between mechanisms may be possible while also leveraging and contrasting biological data to RNNs trained to perform this task. In fact, theoretical work suggests that flexible and cross-context computations rely on the re-purposing of modular dynamical motifs⁴⁹, and our group has recently demonstrated that modular RNNs more readily allow agents to generalize behavior within this naturalistic, navigate-to-target task²². At first approximation, here we may speculate that in the macaque the algorithmic modularity^{22, 46, 49} needed for flexible computations in closed-loop behaviors resides in dIPFC, with MSTd more rigidly reflecting the sensory environment and 7a more closely associated with the motoric aspects of the task and perhaps “reading” from dIPFC.”

Additional comments:

(4) “The slope of this correlation quantifies a change in the gain of neural responses driven by task-relevant stimuli”. This is true only if the tuning stays the same? If the tuning changes, this measure can be zero, even though the gain has changed a lot.

The reviewer is correct. We have now added substantial evidence (**Fig. S7B, S7C, S9C, S6D**) that the tuning functions stay the same (apart from a gain-modulation in MSTd). We have also re-written this section of the text for clarity. The text reads:

*“Given this tuning stability, to quantify the impact of the different optic flow densities on the gain of neural tuning, we examined the gain of the linear regression between tuning functions in different densities of neurons stably tuned ($r > 0.5$, 95% of units; see **Fig. S7B** for example regressions). While we observed significant gain modulation in all brain regions (all $p < 0.005$), the effect size was considerable in MSTd (Cohen’s $d = 0.21$) and negligible in the other areas (Cohen’s $d < 0.06$ in 7a and dlPFC, **Fig. 2B**; examples in raw data in **Fig. S5**, and at the level of tuning functions in **Fig. S7B**).”*

(5) “While the location of mean endpoints did not change across densities”. Not clear how this relates to the observation in Fig. 1C that the radial bias seems to change across contexts. Would a change in bias not result in a change of the means?

The radial and angular biases do not change significantly across density conditions:

*“Task performance was first quantified by expressing the monkeys’ trajectory endpoints and target locations in polar coordinates, with an eccentricity from straight-ahead (θ) and a radial distance (r , **Fig. 1A**, rightmost). **Figure 1B** shows radial (top; r vs. \tilde{r} gain = 0.90; $R^2 = 0.55$) and angular (bottom; θ vs. $\tilde{\theta}$ gain = 0.95; $R^2 = 0.78$) responses as a function of target location for an example session (linear regression $\tilde{r} \sim \text{gain} \times r + \text{intercept}$, see Methods and REF^{25-28, 35, 37}). Most importantly, across all animals ($n = 3$) and sessions ($n = 82$), macaques were accurate in navigating to targets, both during high- (gains, mean \pm S.E.M., radial = 0.85 ± 0.04 ; angular = 0.82 ± 0.14) and low-density conditions (radial = 0.79 ± 0.04 ; angular = 0.77 ± 0.12 ; contrast of high vs. low density separately for each animal and displacement dimension, all $p > 0.13$, **Fig. 1C**, see Behavioral Analyses in Methods). Thus, macaques were able to adaptively estimate their evolving distance to target regardless of the density of optic flow elements and the required computation.”*

Reviewer #3 (Remarks to the Author):

Noel, Balzani, and colleagues.

This is an interesting paper that contains a large amount of new data drawn from three cortical regions – in parietal, prefrontal, and extrastriate visual cortex – in the macaque as they adjust to changes in sensory context. Changes in interneuronal coupling patterns in the dlPLFC and 7a occur as the sensory context changes and in dlPFC these changes are predictive of a stable population code and stable behaviour. I think that the results will be of broad interest. However, the brevity of the format means that sometimes some points are difficult to understand and some additional clarification would be helpful.

1. Could you clarify the angular bias and radial bias indices discussed on page 5 and illustrated in figure 1c? What is the bias towards here? The mean angular and radial position across trials? Or something else (such as the last trial). Do these perhaps relate to the section on page 16 that says “A linear model with multiplicative gain accounted well for the observed data (average $R^2 = 0.72$). Thus, we used the slopes of the corresponding linear regressions as a measure of bias” ? If so, maybe the reader could be guided to this section of the Methods?” What slopes are being taken from the linear regression? What are they related to? It is not clear why they constitute a measure of bias; could this be explained?

We thank the reviewer for this comment and apologize for the lack of clarity. Indeed, by “bias” we meant the gain in a linear regression of the type: response = gain x target location + intercept. These gains, being smaller than 1 indicate under-shooting. Values above 1 would have suggested over-shooting (see REFS^{25-28, 35, 37} for a similar approach). We have amended the manuscript throughout to avoid the term “bias”, given that as pointed out by the reviewer this could be confusing and suggesting a bias either away or toward another variable. The responses are not biased by the previous target location or response location (linear regression including these variables did not improve fit quality; $p > 0.14$). The manuscript has been modified in the following manner;

Results:

“Task performance was first quantified by expressing the monkeys’ trajectory endpoints and target locations in polar coordinates, with an eccentricity from straight-ahead (θ) and a radial distance (r , **Fig. 1A**, rightmost). **Figure 1B** shows radial (top; r vs. \tilde{r} gain = 0.90; $R^2 = 0.55$) and angular (bottom; θ vs. $\tilde{\theta}$ gain = 0.95; $R^2 = 0.78$) responses as a function of target location for an example session (linear regression $\tilde{r} \sim \text{gain} \times r + \text{intercept}$, see Methods and REF^{25-28, 35, 37}). Most importantly, across all animals ($n = 3$) and sessions ($n = 82$), macaques were accurate in navigating to targets, both during high- (gains, mean \pm S.E.M., radial = 0.85 ± 0.04 ; angular = 0.82 ± 0.14) and low-density conditions (radial = 0.79 ± 0.04 ; angular = 0.77 ± 0.12 ; contrast of high vs. low density separately for each animal and displacement dimension, all $p > 0.13$, **Fig. 1C**, see Behavioral Analyses in Methods). Thus, macaques were able to adaptively estimate their evolving distance to target regardless of the density of optic flow elements and the required computation.”

Caption Figure 1:

“**B. Example session.** Monkeys tended to undershoot targets in radial distance (top) and eccentricity (bottom). For each session, we fit a linear regression (response = gain x target location) and thus a gain of ‘1’ indicates no bias. Gains smaller than 1 indicate undershooting.”

Methods:

“**Behavioral Analyses.** The location of targets and the monkey’s end locations were expressed in polar coordinates, with a radial distance (target = r , response = \tilde{r}) and eccentricity from straight ahead (target = θ ; response = $\tilde{\theta}$). On a subset of trials (~13%) animals stopped after <0.5m, suggesting they aborted the trial. Similarly, on a subset of trials (~5%) animals did not stop during the course of a trial (max duration = 7 seconds). These trials were discarded before further behavioral analyses. A linear model with multiplicative gain (i.e., response = gain x target location + intercept; see^{25-28, 37} for the same approach) accounted well for the observed data (average $R^2 = 0.72$). Thus, we used the gain term of the corresponding linear regressions as a measure of

behavioral performance (gain = 1 would indicate no over- or under-shooting of targets, gains < 1 indicate undershooting, and gains > 1 indicate overshooting. The average intercept = 1.1cm in radial distance and 0.2 deg, neither significantly different from 0, both $p > 0.81$; Fig. 1B and C). We also report the fraction of trials in which the animal was rewarded (reward radius = 65cm centered on the firefly, determined by staircase such that animals would be rewarded on ~66% of trials). Lastly, to index variance in endpoint responses we compute for each session the S.E.M of distances to target (rendering a two-dimension error, along x and y, into a single dimension, distance). We then report the mean S.E.M and variance across sessions and density conditions. “

2.1. It is difficult to reconcile figure 2d and the text describing it at the top of page 7. The top of page 7 describes MSTd has showing the least remapping of inter-unit correlations but the figure 2d shows the green line as the line closest to the “maximum” line. Some more explanation of what is being computed and what is being illustrated would be helpful.

The x-axis in **Figure 2D** are correlation coefficients between coupling filters in high and low optic flow density conditions (same animal, session, and sender and receiver units). The y-axis is a cumulative density functions, or cumulative probabilities. These are akin to histograms in that they show the full distribution. We plot the data in this fashion as opposed to histograms given the heavy rightward-tail of the data (i.e., correlations are very high in general). The maximum correlations we would expect (given noise levels in the recordings and the statistical methods) were determined by correlating coupling filters (same animal, session, and sender and receiver units) as estimated in odd vs. even trials. Thus, MSTd (green) being closest to “max” indicates that the coupling filters in this area were the most stable, or with least remapping (i.e., with the highest correlation across high and low optic flow density conditions). We have amended the caption of **Figure 2** to clarify these points. The text reads:

“D. Coupling filter shape stability. Cumulative density functions (i.e., cumulative probability) of the coupling filters correlation coefficients (r) between low- and high-density conditions for all three brain regions (see Fig. S9 for the same data plotted as boxplot). Chance and ceiling-levels were determined by permutation, respectively correlating coupling filters recorded on separate sessions, channels, and animals (i.e., chance level), as well as correlating the same coupling pair (same session, animal, density, sender and receiver units) on odd and even trials (i.e., maximum level). MSTd (green) being the closest to “max” indicates that this region had the most stable (i.e., most correlated) coupling filters across density conditions.”

Further, we have included a new figure (**Fig. S9**) wherein we plot the same data in a more traditional fashion (though not showing the full distributions):

Figure S9. An alternative encoding model reproduces the dynamic remapping of noise correlations in 7a and dIPFC. **A.** Alternative encoding scheme (contrast with Fig. 2A, differences highlighted in orange here). This alternative P-GAM expressed distances to target in egocentric coordinates. Namely, instead of conceiving the targets in a three-dimensional space with depth, the targets are now projected onto the two-dimensional coordinates of a screen. Instead of a radial and angular distance to target, now the targets are expressed by a vertical and horizontal angle vis-à-vis the ongoing location of gaze (also on the screen). Further, we test the hypothesis that animals may not path integrate distances from origin, but from their eventual stopping locations, and thus we include egocentric distances (vertical and horizontal) to the eventual stopping location. Lastly, we include not only eye positions, but also eye velocity. **B.** Fraction of neurons tuned to the different variables in the alternative encoding model. These are congruent with prior results (REF³⁵ and Fig. S7A), showing a patterned mixed selectivity, with greater coding for sensorimotor variables in 7a and of latent variables of dIPFC. Further, the results show greatest coding of eye velocity in MSTd. **C.** Signal correlations (r-value between tuning functions estimated in the different encoding models) are a function of brain area (MSTd in green, 7a in blue, and dIPFC in red) and task variable (restricted to the subset of variables present in both the main-text encoding model and the alternative one presented here). **D.** Coupling filter correlation (r-value in shape of coupling function in low- and high-density optic flow condition) as a function of brain area (MSTd in green, 7a in blue, and dIPFC in red) and encoding schema (main text on the left, and alternative P-GAM on the right). The findings replicate the results demonstrating greater remapping in 7a and dIPFC as opposed to MSTd with the change of optic-flow density. Error bars are ± 1 S.E.M.

2.2. It would be especially helpful to have a sense of what the “cumulative density” is of and how it is calculated. I think that the word “density” here is not related in a simple way to the low density optic flow and high density optic flow conditions. Is that correct? Why are the data being plotted in this way? Given the importance of this figure to the authors’ claims, it would be useful to have a more intuitive explanation of what is being shown here.

Indeed, “density” along the y-axis of **Figure 2D** does not relate to optic flow density. We partially addressed this question in the response to Question 2.1., and maintain the original visualization in the main text in order to give readers a sense of the variance (full distributions). However, in response to this question we have also added a supplemental figure (**Figure S9**) where we visualize the correlation between coupling filters across optic flow density conditions by plotting means and standard errors of the mean.

The figure is presented in response to the precedent question.

3. Is it possible to show readers where the recording sites were? Or, if that is not possible, could more verbal description be given? While the MSTd might be a relatively well defined region, the borders of some of the others are treated differently by different researchers. For example, dIPFC is used to refer to quite a wide region of tissue from the periarculate area to considerably further forward, from within principal sulcus to dorsal to dorsal convexity.

Yes, we apologize for the omission of this important information in the original version of the manuscript. We have added a supplemental figure (**Figure S4**) showing the location of Utah arrays and insertions of linear probes. The figure and caption are the following:

Figure S4. Images and reconstruction of recording sites. **A.** Pictures showing the location of Utah array implants in Monkey S (orange, top = dorso-lateral pre-frontal cortex, bottom = area 7a) and Monkey Q (area 7a). **B-E.** Magnetic resonance imaging (MRI) reconstruction. The rendering of the brain is from Monkey S, and shows all recording sites (arrays and linear probes) on this common reference. Location of acute recordings with linear probes are indicated by spheres, color coded by monkey (S in orange, Q in purple, and M in cyan). Location of Utah arrays are indicated by squares, also color coded by monkey. Brain areas are also indicated by color (dIPFC in red, area 7a in blue, and MSTd in green). MSTd is directly ventral to 7a and shown on the surface here for illustration). AS, arcuate sulcus; IPS, intraparietal sulcus; PS, principal sulcus; STS, superior temporal sulcus; LF, lateral fissure. Figure adapted from REF³⁵.

Further, we have amended the text:

*“We recorded single unit spiking activity simultaneously in sensory (i.e., dorsomedial superior temporal area, MSTd, 240 neurons), parietal (area 7a, 2647 neurons) and prefrontal cortices (dorsolateral prefrontal cortex, dIPFC, 445 neurons; see **Fig. S4** for images and MRI reconstruction of recording sites) to probe how cortical nodes, and in particular their statistical dependencies, support a population code and adaptive beliefs despite changing environments.”*

P3 “sufficient to identity...” should probably be “sufficient to identify...”

The typo has been corrected, thank you.

P11 typo in “input to the system are state-dependent and continuously changing”.

Thank you, this has been corrected.

Reviewer #1 (Remarks to the Author):

I commend the authors for one of the most thorough responses I have ever seen. It answered all my questions more than adequately. I have only one further suggestion that I think would improve the paper. To my mind, one of the most important results is in the correlations with behavioral performance, described in the paragraph beginning on line 403. The authors consider it of sufficient importance to be worth mentioning in the Abstract, so I suggest that it get a figure. Not strictly necessary, since it is well-captured by the r values, but there's a category of reader who decides what is important in a paper by what is in the figures.

Reviewer #2 (Remarks to the Author):

The authors did an excellent job in addressing my comments. I have no further comments.

Reviewer #3 (Remarks to the Author):

The authors have responded to all my requests for clarifications and I think that the manuscript will now be much easier to read.

Reviewer #1 (Remarks to the Author):

I commend the authors for one of the most thorough responses I have ever seen. It answered all my questions more than adequately. I have only one further suggestion that I think would improve the paper. To my mind, one of the most important results is in the correlations with behavioral performance, described in the paragraph beginning on line 403. The authors consider it of sufficient importance to be worth mentioning in the Abstract, so I suggest that it get a figure. Not strictly necessary, since it is well-captured by the r values, but there's a category of reader who decides what is important in a paper by what is in the figures.

We thank the reviewer for his or her positive feedback. We agree the correlation with behavioral performance is one of the most important findings. This correlation was already presented as **Figure 3F** in the revision. Thank you for helping us strengthen the manuscript.

Reviewer #2 (Remarks to the Author):

The authors did an excellent job in addressing my comments. I have no further comments.

Thank you for the constructive feedback which has improved the manuscript.

Reviewer #3 (Remarks to the Author):

The authors have responded to all my requests for clarifications and I think that the manuscript will now be much easier to read.

Thank you for the constructive feedback which has improved the manuscript and increased its clarity.